# PRESERVING FORGERY ARTIFACTS: AI-GENERATED VIDEO DETECTION AT NATIVE SCALE

**Zhengcen Li[1,2], Chenyang Jiang[1,2], Hang Zhao[1], Shiyang Zhou[1], Yunyang Mo[1]**
**Feng Gao[3], Fan Yang[3], Qiben Shan[2], Shaocong Wu[2†], Jingyong Su[1†]**

[1]Harbin Institute of Technology, Shenzhen [2]Peng Cheng Laboratory [3]Peking University
wushc@pcl.ac.cn   sujingyong@hit.edu.cn

Project Page: https://github.com/mgiant/Qwen2.5ViT-AIGVDetection

## ABSTRACT

The rapid advancement of video generation models has enabled the creation of highly realistic synthetic media, raising significant societal concerns regarding the spread of misinformation. However, current detection methods suffer from critical limitations. They rely on preprocessing operations like fixed-resolution resizing and cropping. These operations not only discard subtle, high-frequency forgery traces but also cause spatial distortion and significant information loss. Furthermore, existing methods are often trained and evaluated on outdated datasets that fail to capture the sophistication of modern generative models. To address these challenges, we introduce a comprehensive dataset and a novel detection framework. First, we curate a large-scale dataset of over 140K videos from 15 state-of-the-art open-source and commercial generators, along with Magic Videos benchmark designed specifically for evaluating ultra-realistic synthetic content. In addition, we propose a novel detection framework built on the Qwen2.5-VL Vision Transformer, which operates natively at variable spatial resolutions and temporal durations. This native-scale approach effectively preserves the high-frequency artifacts and spatiotemporal inconsistencies typically lost during conventional preprocessing. Extensive experiments demonstrate that our method achieves superior performance across multiple benchmarks, underscoring the critical importance of native-scale processing and establishing a robust new baseline for AI-generated video detection.

## 1   INTRODUCTION

Artificial Intelligence-Generated Content (AIGC) has advanced rapidly, revolutionizing the creation of high-quality text (Yang et al., 2024; DeepSeek-AI, 2024), image (Esser et al., 2024; Labs, 2024), audio (Kreuk et al., 2023; Copet et al., 2023) and video (Brooks et al., 2024). Among these advancements, video generation has seen particularly significant progress, evolving from foundational models like Stable Diffusion (Rombach et al., 2022) to more advanced architectures such as Diffusion Transformers (DiTs) (Peebles & Xie, 2023; Brooks et al., 2024), as well as proprietary commercial products (Pika Labs, 2023; Jimeng AI, 2024; Kuaishou, 2024). These developments have pushed the boundaries of deepfake technologies (Yang et al., 2022), enabling large-scale creation of fully AI-generated videos. However, the emergence of near-photorealistic synthetic videos poses serious threats to privacy, reputation, and public trust (Wang et al., 2024), underscoring the urgent need for effective detection and mitigation strategies against disinformation and misinformation.

Deepfake detection (Yan et al., 2023) and AI-generated image detection (Wang et al., 2020; Zhu et al., 2023) have made significant progress in identifying manipulated content. However, existing deepfake detection methods (Qian et al., 2020; Xu et al., 2023; Oorloff et al., 2024; Nguyen et al., 2024) often face generalizability issues as they primarily focus on detecting facial forgeries. Meanwhile, approaches for detecting images generated by Generative Adversarial Networks (GAN) and

---

[†]Corresponding author.

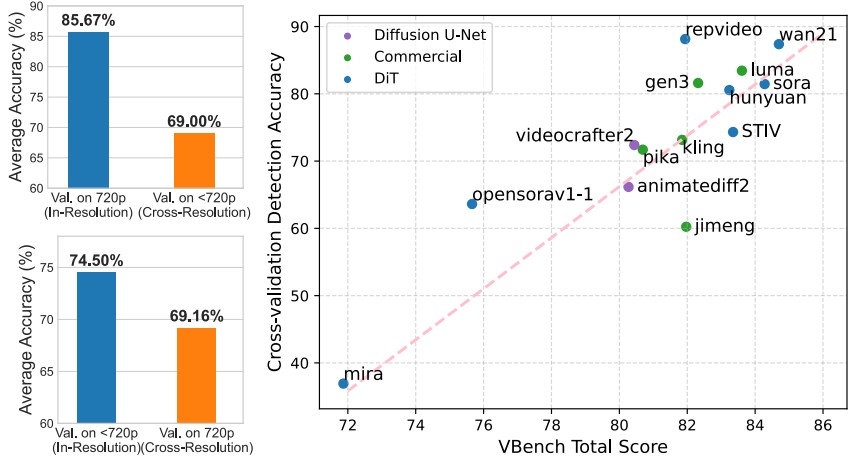

Figure 1: **Resolution mismatch and generator quality strongly affect cross-generator video detection. Left:** Detectors trained on 720p videos (top) and on lower-resolution videos (<720p; bottom) both exhibit a pronounced performance drop when evaluated at a spatial resolution different from that used during training. **Right:** We observe a strong positive correlation between generator quality (VBench score) and cross-validation performance (Pearson $\rho = 0.86$), indicating that higher-quality generators tend to yield more transferable training data for detector learning. These findings motivate a unified framework that is robust to resolution shifts and generator-specific artifacts.

diffusion models (Wang et al., 2020; 2023c; Tan et al., 2024; Luo et al., 2024) are typically restricted to static media, leaving general spatiotemporal forgery detection largely unaddressed.

Recent studies have begun to develop more robust solutions for AI-generated image and video detection (Yan et al., 2025; Li et al., 2025; Song et al., 2024; Chen et al., 2024b; Kundu et al., 2025b). A significant and shared limitation among these methods is the conventional preprocessing of resizing (Yan et al., 2025) or cropping (Li et al., 2025) input frames to a fixed resolution (e.g., 224x224). Forgery detection methods often rely on two types of features, subtle artifacts and high-level semantics (Cheng et al., 2025). However, this fixed-resolution preprocessing degrades both types of features. Resizing distorts the original aspect ratio, misleading detectors into learning superficial distribution differences rather than robust and generalizable forgery features (Rajan et al., 2025). Cropping, meanwhile, can discard important content outside the selected area, thereby discarding global semantic cues of high-resolution content. Furthermore, both downsampling approaches degrade the subtle, pixel-level artifacts that are critical for identifying synthetic media and capturing fine-grained inconsistencies (Corvi et al., 2025).

Furthermore, progress in AI-generated video detection is hampered by the use of outdated synthetic data sources. Existing datasets (Chen et al., 2024b; Song et al., 2024) are predominantly composed of videos generated by earlier models, which typically exhibit low resolution, limited quality, and short durations. As a result, detection models trained on these datasets experience a significant performance drop when evaluated on modern AI-generated videos. To better understand these challenges, we conduct cross-validation experiments using existing detectors on a synthetic videos dataset sourced from 14 generative models. Our preliminary results reveal two critical insights, as illustrated in Figure 1. First, we observe a significant performance drop when detectors are evaluated on videos with different resolution from those in the training set. Second, detection performance is positively correlated with the quality of the video generators, meaning that stronger detectors require training on higher-quality, more realistic synthetic videos. These findings further highlight the importance of constructing a high-quality and diverse dataset, as well as a training framework capable of effectively handling videos with diverse resolutions, durations and generative sources.

In response to the limitations of existing methods, we propose a unified framework that supports training and evaluation on videos with diverse resolutions and generative sources. First, we curate a high-quality and diverse video dataset sourced from 15 representative video generation models for training and develop a meticulously crafted pipeline to synthesize high-quality, human-indistinguishable videos for evaluation, termed Magic Videos. Second, we design a native-resolution

training framework based on the Qwen2.5-VL Vision Transformer (Bai et al., 2025) (Qwen2.5-ViT), which unifies image and video modeling and enables the model to natively process videos with arbitrary spatial resolutions and temporal lengths. By removing the constraints of fixed-size downsampling preprocessing, our method achieves strong generalization capabilities to capture general spatiotemporal forgery artifacts. Extensive experiments on a wide range of benchmarks (Genvideo (Chen et al., 2024b), DVF (Song et al., 2024) and our proposed Magic Videos) demonstrate that our model is robust and achieves state-of-the-art performance in detecting AI-generated videos. Our Contributions are summarized as follows:

- We introduce a new high-quality diverse dataset sourcing from 15 generators for training, and curate Magic Videos with 6 recent generators for evaluation, ensuring that both training and evaluation are aligned with the current generative quality of AIGC.

- We propose a novel native-resolution framework built upon the Qwen2.5-ViT, which processes videos with native aspect ratio and variable resolution, preserving crucial forgery artifacts often lost during conventional resizing or cropping.

- Through extensive experiments, we demonstrate that our method achieves state-of-the-art performance and robust generalization across a wide range of benchmarks, setting a new standard for AI-generated video detection.

## 2 RELATED WORK

### 2.1 VIDEO GENERATIVE MODELS

Diffusion models (Ho et al., 2020; Song et al., 2022; Rombach et al., 2022) have significantly enhanced the quality and controllability of image generation, inspiring researchers to extend these techniques to video generation tasks. Early work (Singer et al., 2022) propose incorporating motion dynamics into pre-trained text-to-image generation models. More recent studies (Chen et al., 2024a; Guo et al., 2024; Blattmann et al., 2023; Wang et al., 2023a; Wei et al., 2024) leverage latent-based diffusion models (Rombach et al., 2022) to generate short dynamic videos from text or image inputs. With the growing popularity of Diffusion Transformers (DiTs) (Peebles & Xie, 2023) in image generation (Labs, 2024), DiT and its variants (Esser et al., 2024) have been widely proposed for video generation tasks (Ma et al., 2024b; Zheng et al., 2024; Brooks et al., 2024; Yang et al., 2025; Kong et al., 2024; Wan Team, 2025; Polyak et al., 2024). Besides Diffusion based methods, Generative Adversarial Networks (GANs) (Shen et al., 2023; Wang et al., 2023b) are also explored for video generation. The success of decoder-only architecture in language model has also motivated research in generating long videos using autoregressive models (Kondratyuk et al., 2024; Yu et al., 2023; Yin et al., 2025). Commercial video generation products (Brooks et al., 2024; Kuaishou, 2024; Jimeng AI, 2024; Pika Labs, 2023; MiniMax, 2024), employ complex and proprietary pipelines and produces hyper-realistic videos. However, the lack of transparency surrounding these systems limits detailed analysis of their methodologies.

In this paper, we propose a generative video dataset that encompasses most of the aforementioned architectures, including Diffusion U-Net (Chen et al., 2024a; Guo et al., 2024), DiT (Brooks et al., 2024; Wan Team, 2025; Ju et al., 2024; Zheng et al., 2024; Polyak et al., 2024; Lin et al., 2024; Ma et al., 2025), MMDiT (Kong et al., 2024; Si et al., 2025), and auto-regressive (Yin et al., 2025) models. The diversity of generative models included in our dataset ensures broad coverage and supports the generalizability of the proposed method.

### 2.2 AI-GENERATED IMAGE AND VIDEO DETECTION

**AI-Generated Image Detection.** As generative technologies rapidly advance, a growing number of forged images are now entirely synthesized by GANs (Goodfellow et al., 2014) and Diffusion models (Rombach et al., 2022), moving beyond traditional limited manipulation techniques. Consequently, substantial research efforts have focused on developing generalizable synthetic image detection methods (Tan et al., 2023; Ojha et al., 2023; Yan et al., 2024; Liu et al., 2024b), including approaches based on reconstruction error (Wang et al., 2023c; Luo et al., 2024; Guillaro et al., 2025), pixel-level features (Wang et al., 2020; Tan et al., 2024; Cheng et al., 2025), or adapting visual backbones (Koutlis & Papadopoulos, 2024; Yan et al., 2025; Liu et al., 2024a). These methods

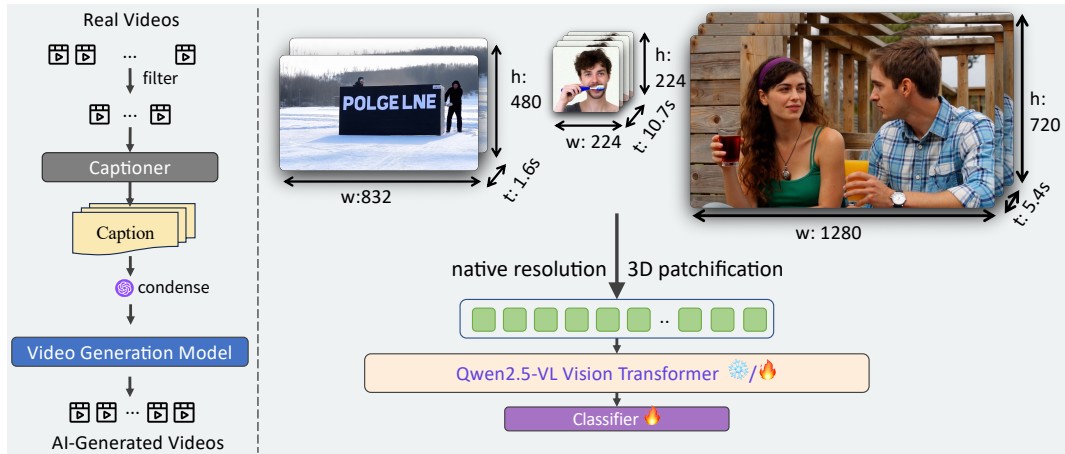

Figure 2: **Overview of the data generation pipeline and the proposed detection framework.**
**Left:** We curate high-quality captions from real videos and refine them into prompts for state-of-the-art text-to-video generators, producing realistic synthetic videos for training and evaluation.
**Right:** Our detector supports variable spatial resolutions and temporal lengths. It avoids fixed-size resizing/cropping and applies 3D patchification to preserve the input aspect ratio and fine-grained, high-frequency forensic cues that are often weakened by conventional downsampling. Built on the Qwen2.5-VL Vision Transformer, the framework models videos as sequences of spatiotemporal patches for robust AI-generated video detection.

are typically trained on images generated by specific models (Karras et al., 2018; Song et al., 2022) and aim to achieve cross-architecture generalization.

**AI-Generated Video Detection.** More recently, research has expanded to the detection of fully AI-generated videos (Ma et al., 2024a; Ni et al., 2024; Ji et al., 2024; Chang et al., 2025). VLM-based methods (Song et al., 2024; Wen et al., 2025) prompt large vision-language models to identify unnatural AI-like cues, while ViT-based methods (Chen et al., 2024b; Corvi et al., 2025) introduce forgery-posed generated datasets and design modules to detect spatial-temporal inconsistencies. However, existing methods for detecting AI-generated images and videos commonly suffer from a reliance on fixed resizing operations. Such preprocessing can lead to the loss of fine-grained details and spatial distortions, ultimately compromising model robustness across diverse inputs. In this work, we address this issue by training on native spatial resolution and temporal duration, without resizing or temporal padding. This design fundamentally avoids the pitfalls of conventional preprocessing and significantly enhances the model's generalization capability.

## 3 METHODOLOGY

### 3.1 DATA CURATION

**Selection of Data Sources.** The AI-generated videos are curated from multiple sources: (1) VBench (Huang et al., 2023; 2024), which provides generated videos from various text-to-video models using a predefined suite of diverse prompts; (2) Movie Gen (Polyak et al., 2024), which contributes videos generated by its proprietary model; and (3) A collection of highly realistic videos synthesized using various cutting-edge open-source and commercial models, guided by our custom-designed prompt library.

**Realistic Video Generation Pipeline.** To evaluate the capability of generative content detectors in real-world scenarios, we design a pipeline for constructing synthetic videos that closely resemble authentic content. We prioritize scenarios that pose significant risks to information security, such as realistic landscapes, architectural scenes, and human interactions, as these categories are particularly susceptible to misuse and misinformation due to their inherent plausibility. Leveraging ShareGPT4Video (Chen et al., 2024c) repository of detailed and high-quality captions, we curate

| Data | Source | Number | Resolution | Duration |
|---|---|---|---|---|
| Training Data (15 models) | Vbench | 140K | 240p-768p | 1-10s |
| Movie Gen | MovieGenBench | 2003 | 1920x1088 | 10.7s |
| Wan2.1 | Open Source | 450 | 1280x720 | 5.4s |
| Wan-1.3B | Open Source | 584 | 832x480 | 5s |
| Hailuo | API (T2V-01) | 450 | 1280x720 | 5.6s |
| Seaweed | API (Jimeng-S2.0) | 450 | 1472x832 | 5s |
| Seedance | API (Jimeng-S3.0) | 450 | 1248x704 | 5s |
| StepVideo | Open Source | 450 | 950x540 | 8.2s |

Table 1: **Dataset statistics for training/validation and the Magic Videos benchmark.** See appendix for details of these datasets.

content specifically within these realism-oriented themes. To accommodate the capabilities of state-of-the-art architectures, we filter videos by duration (3-12 seconds) and caption length (fewer than 1000 characters). The curated prompts are further optimized using GPT-4o to condense the description to under 500 characters. Table 1 summarizes videos that are synthesized by six distinct video generators using our comprehensive prompt library. These videos represent the current frontier of photorealistic synthetic content, enabling a rigorous assessment of detection models under practical and high-risk conditions.

## 3.2 QWEN2.5-VIT

Contemporary AI-generated content detectors primarily operate by identifying two categories of features: local artifacts and global semantic inconsistencies (Cheng et al., 2025). However, a common practice in existing methodologies is to resize input images to a low, fixed resolution, typically 224x224 pixels. This downscaling operation adversely affects the features crucial for detection: it degrades subtle local artifacts and distorts global semantic structures. In this paper, we introduce a unified framework that processes images and videos at native resolution, thereby preserving the original forgery artifacts. The framework begins by tokenizing input videos into 3D patches at the native scale and adopts Qwen2.5-ViT (Bai et al., 2025) as a novel visual backbone for general video forgery detection.

**3D Video Patchifying at Native Scale.** We follow the video processing steps of (Bai et al., 2025), which introduces a 3D patch partitioning strategy that enables native-resolution inputs. For static images, it employs a standard spatial patch extraction method (e.g., 14x14 pixels). Unlike conventional ViTs that operate on static frames independently, our method extends patchification into the temporal dimension for video data. Given an input video tensor $V \in \mathbb{R}^{T \times H \times W \times C}$, it partitions $V$ to non-overlapping 3D patches of size $(P_t, P_h, P_w) = (2, 14, 14)$ and computes patch embedding via linear projection matrix $E$. This design eliminates the need for conventional resizing and padding operations, allowing the Transformer model to operate natively on both the spatial and temporal scales. The initial transformation of a raw video tensor $V$ into a sequence of feature embeddings $X^{(0)}$ is described in Equation (1). The 3D patchification is particularly effective in detecting subtle texture artifacts and temporal inconsistencies at the patch level. By preserving the original resolution during preprocessing, our method ensures that potential features critical for forgery detection remain intact and undistorted.

$$X^{(0)} = \text{Unfold}(V; P_t, P_h, P_w)^T \cdot E \tag{1}$$

**Transformer Layer Structure.** Qwen2.5-ViT consists of 32 Transformer layers, each adopting a pre-normalization structure, in which RMSNorm is applied before both the self-attention and feed-forward network (FFN). The FFN component employs the SwiGLU activation function. To effectively encode the spatial relationships between patches, 2D Rotary Positional Embedding (RoPE) (Su et al., 2023) is applied to the queries and keys in self-attention, enhancing the model's extrapolation capability across input resolutions. The computations performed within each Trans-

former layer are described as

$$\hat{X}^{(l)} = X^{(l-1)} + \text{Attention}(\text{RMSNorm}(X^{(l-1)})),$$
$$X^{(l)} = \hat{X}^{(l)} + \text{FFN}_{\text{SwiGLU}}(\text{RMSNorm}(\hat{X}^{(l)})). \tag{2}$$

In Equation (2), $X^{(l-1)}$ and $X^{(l)}$ denote the input and output hidden states of the $l$-th Transformer layer, respectively.

**Infrastructure Optimization for Efficiency.** To address the computational challenges associated with high-resolution inputs, which typically lead to quadratic complexity, several optimizations are integrated. A batch packing strategy from NaViT (Dehghani et al., 2023) is adopted to allow the model to handle variable-length sequences without padding or attention masks. This is combined with Flash Attention (Dao, 2023), enabling GPU awareness of sequence boundaries and significantly improving both computational efficiency and memory usage through optimized CUDA kernels. In addition, a hybrid attention strategy is adopted where the majority of Transformer layers utilize $114 \times 114$ windowed attention, ensuring that the computational cost scales linearly with the number of input patches.

**Classifier and Tuning Methods.** For the final binary classification task of distinguishing between authentic and AI-generated content, we append a simple yet effective classification head to the Qwen2.5-ViT backbone. The output tokens from the final Transformer layer is first aggregated into a single, fixed-size feature vector using global average pooling. This vector is then passed through a single fully connected (FC) linear layer that outputs the logits corresponding to the "real" and "generated" classes. To adapt the pre-trained model to this task, we explore three fine-tuning strategies: (1) Full Finetuning: Both the visual backbone and classification head are jointly optimized during training. (2) Linear-Probing: Serves as a baseline, where the entire vision backbone is frozen and only the classification head is trained. (3) Parameter-Efficient Fine-Tuning (PEFT): Specifically. we adopt Low-Rank Adaptation (LoRA (Hu et al., 2021)), which introduces small, trainable low-rank matrices into the frozen backbone, allowing only a subset of parameters to be updated.

## 4 EXPERIMENTS

### 4.1 DATASETS

**Training Dataset.** We construct a training set of 70K AI-generated videos and 70K real videos. The synthetic videos are generated by VBench (Huang et al., 2023) using their prompt set, while the real ones are sampled from MSVD (Chen & Dolan, 2011) and Kinetics (Kay et al., 2017). For validation, we use 1,003 fake videos from MovieGenVideoBench (Polyak et al., 2024) and 1,000 real videos from Panda-70M (Chen et al., 2024d).

| Model | Training Data | Movie Gen | Wan 2.1 | Wan-1.3B | Hailuo | Seaweed | Seedance | StepVideo | mACC |
|---|---|---|---|---|---|---|---|---|---|
| RINE† | ldm | 52.97 | 45.35 | 33.56 | 51.4 | 51.63 | 44.65 | 70.23 | 49.47 |
| FatFormer† | ProGAN | 50.02 | 50.00 | 50.17 | 50.00 | 50.00 | 50.00 | 50.23 | 50.07 |
| B-Free† | SD 2.1 | 64.30 | 53.72 | 68.32 | 55.81 | 31.63 | 36.05 | 48.60 | 49.02 |
| Effort† | SD 1.4 | 70.74 | 73.26 | 66.95 | 67.44 | 86.05 | 83.26 | 62.79 | 73.29 |
| WaveRep† | Pyramid Flow | 65.30 | 58.84 | 53.6 | 59.53 | 59.53 | 57.21 | 59.53 | 58.04 |
| F3Net | | 92.51 | 71.86 | 69.86 | 66.98 | 74.88 | 73.26 | 66.98 | 70.64 |
| TALL | | 91.71 | 64.65 | 67.81 | 61.86 | 64.42 | 63.49 | 63.49 | 64.29 |
| NPR | | 92.66 | 71.63 | 66.27 | 72.33 | 74.88 | 73.26 | 72.09 | 71.74 |
| TimeSformer | 15Model-140K | 91.41 | 68.84 | 72.09 | 66.28 | 69.77 | 67.91 | 67.21 | 68.68 |
| CLIP ViT-L/14 | (Ours) | 99.20 | 78.14 | 77.74 | 77.21 | 77.91 | 77.21 | 75.58 | 77.30 |
| X-CLIP-B/16 | | 98.55 | 76.28 | 72.43 | 75.12 | 75.12 | 75.35 | 72.33 | 74.44 |
| X-CLIP-L/14 | | 98.85 | 80.00 | **86.13** | 79.53 | 78.60 | 80.00 | **79.53** | 80.63 |
| Moon-ViT | | 98.25 | 76.74 | 79.62 | 75.81 | 76.74 | 75.81 | 74.88 | 76.60 |
| Qwen2.5-ViT (Ours) | | 97.20 | **85.81** | 83.39 | **84.65** | **83.95** | **84.88** | 76.51 | **83.20** |

Table 2: **Accuracy (ACC) Benchmarking Performance on on Movie Gen (val) and Magic Videos (test), reported per generator and averaged (mACC).** † Results are produced with the official pretrained model. Best: **bold**; second best: underlined.

| Model | Training Data | Movie Gen | Wan 2.1 | Wan-1.3B | Hailuo | Seaweed | Seedance | StepVideo | mAP |
|---|---|---|---|---|---|---|---|---|---|
| RINE† | ldm | 71.11 | 40.41 | 32.69 | 47.46 | 46.54 | 38.98 | 69.93 | 46.00 |
| FatFormer† | ProGAN | 58.84 | 39.06 | 54.96 | 43.80 | 44.55 | 42.16 | 52.36 | 46.15 |
| B-Free† | SD 2.1 | 70.38 | 59.80 | 73.81 | 62.62 | 36.08 | 38.93 | 52.37 | 53.94 |
| Effort† | SD 1.4 | 80.60 | 76.61 | 73.75 | 70.67 | 95.48 | 91.65 | 63.41 | 78.60 |
| WaveRep† | Pyramid Flow | 92.97 | 83.50 | 85.31 | _95.64_ | _99.12_ | 78.64 | _94.04_ | 89.38 |
| F3Net | | 96.20 | 85.79 | 78.05 | 80.31 | 92.13 | 87.32 | 81.26 | 84.14 |
| TALL | | 96.07 | 83.59 | 80.78 | 78.58 | 88.81 | 82.81 | 83.39 | 82.99 |
| NPR | | 97.10 | 86.93 | 82.70 | 87.51 | 92.71 | 92.08 | 90.96 | 88.82 |
| TimeSformer | 15Model-140K | 96.91 | 82.44 | 83.04 | 75.42 | 83.49 | 80.29 | 78.55 | 80.54 |
| CLIP ViT-L/14 | (Ours) | 99.95 | _98.78_ | 93.14 | 87.43 | 97.58 | 92.05 | 86.85 | 92.64 |
| X-CLIP-B/16 | | 99.87 | 94.94 | _94.07_ | **95.76** | 87.65 | **97.78** | 81.65 | 91.98 |
| X-CLIP-L/14 | | 99.94 | **99.62** | 82.73 | 92.91 | **99.16** | 96.18 | **95.71** | **94.39** |
| Moon-ViT | | 99.24 | 93.84 | 90.39 | 87.24 | 94.11 | 90.68 | 81.69 | 89.66 |
| Qwen2.5-ViT (Ours) | | 99.46 | 96.67 | **99.68** | 94.20 | 91.59 | _96.92_ | 80.63 | _93.28_ |

Table 3: **Average Precision (AP) Benchmarking Performance on on Movie Gen (val) and Magic Videos (test), reported per generator and averaged (mACC)**. † Results are produced with the official pretrained model. Best: **bold**; second best: underlined.

**Test Datasets.** To evaluate robustness against state-of-the-art synthetic videos, we introduce the **Magic Videos** benchmark (Table 1), which consists of high-quality, hyper-realistic videos generated by six cutting-edge video generation models using carefully curated prompts. Each generated video subset is paired with corresponding real videos to support binary classification evaluation. We report performance using Accuracy (ACC) and Average Precision (AP) per generator subset. **External Benchmarks:** In addition, we evaluate our method on the test sets of three external datasets: DVF (Song et al., 2024), GenVideo (Chen et al., 2024b), and DeepTraceReward (Fu et al., 2025). These datasets encompass diverse real-world sources and a wide range of video generation models, enabling a more comprehensive assessment of detector performance across different synthesis techniques and generation stages.

**Baselines.** We benchmark four categories of methods: (1) AI-generated video detection approaches (MM-Det (Song et al., 2024), DeMamba (Chen et al., 2024b), UNITE (Kundu et al., 2025b), TruthLens (Kundu et al., 2025a), and WaveRep (Corvi et al., 2025)); (2) visual and video foundation backbones (X-CLIP-B/16 (Ni et al., 2022), X-CLIP-L/14 (Ni et al., 2022), TimeSformer (Bertasius et al., 2021), and Moon-ViT (Du et al., 2025)); (3) deepfake detection methods (TALL (Xu et al., 2023) and F3Net (Qian et al., 2020)); and (4) general AI-generated image detection methods (NPR (Tan et al., 2024), FatFormer (Liu et al., 2024a), RINE (Koutlis & Papadopoulos, 2024), B-Free (Guillaro et al., 2025), and Effort (Yan et al., 2025)). For image-based methods, we report the results by averaging logits over $T$ frames to obtain video-level predictions.

**Implementation Details.** We train our model for five epochs using the binary cross-entropy loss and the AdamW optimizer. The learning rate is set to $1 \times 10^{-5}$ for full fine-tuning and $1 \times 10^{-4}$ for parameter-efficient fine-tuning (PEFT). To balance performance and computational cost, we adopt the preprocessing strategy described in Bai et al. (2025); Du et al. (2025), which specifies minimum and maximum token budgets for image inputs. Each input frame is resized to the highest possible resolution within the (min_pixels, max_pixels) range while preserving the original aspect ratio. In our experiments, we consider two resolution ranges per frame: $(224 \times 224, 720 \times 720)$ and $(224 \times 224, 448 \times 448)$. For temporal sampling, videos are decoded into frames at 2 fps. During training, we randomly sample $T = 8$ consecutive frames; during evaluation, we instead select the central $T = 8$ frames. Additional implementation details are provided in the Appendix.

## 4.2 AI-GENERATED VIDEO DETECTION

**Evaluation on Magic Videos.** The experimental results presented in Table 2 and Table 3 provide a comprehensive evaluation of our model against several distinct classes of methods. A notable observation is the underwhelming performance of models originally developed for AI-generated image detection, including RINE, FatFormer, B-Free, and Effort. These models exhibit relatively poor performance on video-based benchmarks, even compared to image-based methods that are trained on our video datasets. This discrepancy suggests a fundamental difference between forgery patterns present in static images and those in dynamic video sequences; features learned for detect-

| Method | Video-Crafter | Zero-scope | Open-Sora | Sora | Pika | Stable Diff. | Stable Video | AVG |
|---|---|---|---|---|---|---|---|---|
| CNNDet* | 87.4 | 88.2 | 78.0 | 63.8 | 77.3 | 73.5 | 78.9 | 78.2 |
| DIRE* | 55.9 | 61.8 | 53.8 | 60.5 | 65.8 | 62.7 | 69.9 | 62.1 |
| MM-Det | 93.5 | 94.0 | 88.8 | 86.2 | 95.9 | **95.7** | 89.9 | 92.0 |
| NPR | 86.6 | 85.6 | 96.0 | 81.0 | 94.6 | 71.1 | 97.0 | 87.4 |
| TALL | **95.4** | 91.8 | 97.2 | 94.9 | 97.5 | 83.6 | 98.2 | 92.6 |
| F3Net | 90.4 | 90.2 | 95.9 | 90.1 | 97.8 | 93.1 | 98.5 | 93.7 |
| TimeSformer | 94.5 | 92.7 | 98.0 | 92.5 | 98.4 | 92.4 | 99.5 | 95.4 |
| Qwen2.5-ViT (Ours) | 93.5 | **99.8** | **98.6** | **96.4** | **99.1** | 95.6 | **99.7** | **97.6** |

Table 4: **Benchmarking Results in terms of AUC Performance on DVF-Test Song et al. (2024)**. Results with * are derived from Song et al. (2024).

| Model | Averaged | | | Overall | |
|---|---|---|---|---|---|
| | Recall | F1 | AP | ACC | Recall |
| UNITE | 89.60 | - | 92.76 | - | - |
| TruthLens | - | - | - | 90.49 | - |
| DeMamba-CLIP | **91.58** | 89.19 | 93.45 | 96.14 | 92.29 |
| NPR | 83.01 | 47.99 | 63.66 | 86.75 | 92.40 |
| F3Net | 83.48 | 56.78 | 71.57 | 88.26 | 93.06 |
| TimeSformer | 86.42 | 65.38 | 77.67 | 87.51 | 91.55 |
| TALL | 89.44 | 61.51 | 76.67 | 90.05 | 91.76 |
| CLIP-L | 87.72 | 64.73 | 77.62 | 92.64 | 90.57 |
| XCLIP-B | 89.79 | 53.76 | 72.04 | 92.60 | 90.90 |
| XCLIP-L | 88.94 | 61.60 | 78.84 | 92.74 | 92.43 |
| Qwen2.5-ViT (Ours) | 91.16 | **90.64** | **96.13** | **96.64** | **93.18** |

| Method | ACC | Fake ACC | Real ACC |
|---|---|---|---|
| GPT-5 | 90.7 | 84.6 | 98.8 |
| GPT-4.1 | 92.9 | 89.1 | 97.9 |
| Gemini 2.5 Pro | 84.3 | 75.7 | 95.8 |
| VideoLLaMa3 7B | 10.0 | 38.1 | 4.3 |
| Qwen2.5-VL 7B | 51.7 | 20.2 | 93.4 |
| Qwen2.5-VL 32B | 47.4 | 8.9 | 98.5 |
| Qwen2.5-VL 72B | 50.0 | 16.6 | 94.3 |
| DeepTraceReward (w/ Qwen2.5 VL 7B) | 74.7 | 55.7 | **100.0** |
| Qwen2.5-ViT (Ours) | **97.2** | **96.3** | 98.2 |

Table 5: **Benchmarking Results in terms of averaged Recall, F1, AP per subset and overall Recall and ACC Performance on Genvideo-Val (Chen et al., 2024b)**. Detailed results are in Appendix.

Table 6: **Benchmarking Results in terms of ACC on DeepTraceReward (Fu et al., 2025)**. Results of baseline methods are reported in (Fu et al., 2025).

ing image artifacts do not generalize well to the spatio-temporal domain required for video-level analysis. Similarly, methods designed specifically for deepfake detection, such as F3Net and TALL demonstrate limited effectiveness. While these models excel at identifying at facial manipulations, their specialization becomes a constraint when faced with the broader challenge of detecting fully synthesized videos. In contrast, pretrained visual backbones like TimeSformer, CLIP-ViT and X-CLIP exhibit competitive performance by leveraging extensive pre-training on diverse visual data. However, their effectiveness is ultimately constrained by architectural limitations. A primary issue is the conventional practice of resizing input frames to a fixed resolution of 224×224 pixels. This downsampling process can eliminate subtle forgery artifacts and disrupt global semantic features crucial for detecting sophisticated generative content. Moon-ViT (Du et al., 2025), which applies a similar processing pipeline based on NaViT, also suffers from this limitation as it operates on static images and cannot capture temporal inconsistencies. Our proposed method achieves the highest average scores in both ACC and AP, establishing a new state-of-the-art on these benchmarks. Our proposed method achieves the highest average accuracy (mACC) and highly competitive average precision (mAP), establishing a strong baseline. While our model does not yield the absolute best AP on every individual generator, it consistently delivers strong and balanced performance across all generator types, highlighting its exceptional generalizability. This superior performance is directly attributed to our advanced architecture. By leveraging the Qwen2.5-ViT backbone, our model integrates native-resolution modeling with dynamic temporal duration modeling, thereby avoiding destructive downsampling and preserving the fidelity of forgery cues inherent in the original content. By effectively capturing both fine-grained artifacts and high-level semantic inconsistencies, our model provides a more robust and accurate solution for detecting AI-generated videos.

**Evaluation on DVF-test.** We use the same model weights trained on our 15model-140k dataset to directly perform cross-dataset evaluation on the DVF dataset (Song et al., 2024). The results are

| Archs. | Variants | Magic | Genvideo | Avg. |
|---|---|---|---|---|
| spatial resolution | random crop to 224p | 62.62 | 93.50 | 78.06 |
| | random resize to 224p | 73.69 | 95.52 | 84.61 |
| | dynamic [224p, 448p] | 81.19 | 96.01 | 88.60 |
| | dynamic [224p, 720p] | **83.20** | **96.64** | **89.92** |
| temporal resolution | $T$=2 | 71.15 | 94.70 | 82.93 |
| | $T$=4 | 75.58 | 94.40 | 84.99 |
| | $T$=8 | **81.19** | **96.01** | **88.60** |
| tuning mode | LP | 70.60 | 91.91 | 81.26 |
| | LoRA(r=16) | 78.73 | 94.95 | 86.84 |
| | full | **81.19** | **96.01** | **88.60** |

Table 7: **Ablation studies regarding spatial-temporal resolution and tuning mode.** We report averaged ACC(%) on Magic Videos and Genvideo. For temporal and tuning experiments, the spatial resolution is set to dynamic[224p, 448p].

presented in Table 4. Our model achieves the highest average AUC of 97.6, demonstrating the high quality of our training dataset and the strong generalizability of our model in detecting AI-generated videos across diverse generation techniques.

**Evaluation on GenVideo-Val.** We directly evaluate the model trained on the 15model-140k dataset on GenVideo-Val (Chen et al., 2024b) using the same weights. Owing to the substantial class imbalance between real and generated samples in the GenVideo evaluation subsets, we report both overall recall and accuracy (ACC) for a more comprehensive comparison. As shown in Table 5, our method outperforms all baselines, including UNITE (Kundu et al., 2025b), which is specifically designed for AI-generated video detection, as well as larger MLLM-based TruthLens (Kundu et al., 2025a) and other baseline approaches trained on the same data as ours. Notably, despite using only one-fifteenth of the training data used by DeMamba (Chen et al., 2024b), our model achieves better performance and strong cross-dataset generalization, particularly on videos generated by earlier models.

**Evaluation on DeepTraceReward.** To further demonstrate robustness against unseen generators, we evaluated our method on the DeepTraceReward (Fu et al., 2025), which contains 4,335 videos from 7 recent generators (including Pika-1.5, Kling-1.5, etc). Table 6 compares our Qwen2.5-ViT against leading multimodal LLMs. Our model achieves 97.2% accuracy, significantly outperforming large general-purpose VLMs (e.g., GPT-5, Gemini 2.5 Pro) on the binary classification task. Moreover, while general-purpose VLMs often struggle with detecting synthetic video, our model demonstrates balanced performance (96.3% Fake ACC vs. 98.2% Real ACC), proving its effectiveness in identifying artifacts from the latest generation engines without overfitting to specific training generators.

### 4.3 ABLATION STUDY AND ANALYSIS

We conduct a series of ablation studies, as detailed in Table 7, to systematically investigate the impact of spatial resolution, temporal resolution, and different fine-tuning strategies on our model's performance. We observe that the benefits of high-fidelity inputs are substantially larger on *Magic*. In contrast, GenVideo contains lower-resolution videos with shorter durations; therefore, it is less sensitive to the performance degradation introduced by aggressive downsampling during preprocessing. As a result, improvements brought by higher spatial/temporal fidelity are more pronounced on Magic than on GenVideo.

**Ablation Study on Spatial Resolution.** Our analysis reveals critical performance differences. The conventional random crop to 224p method yields the lowest average accuracy on high-resolution content. Switching to random resize to 224p boosts performance to 73.69, but this approach can still cause degradation of subtle artifacts. In contrast, our dynamic resolution strategy which preserves the original aspect ratio demonstrates markedly superior performance, with the overall average ac-

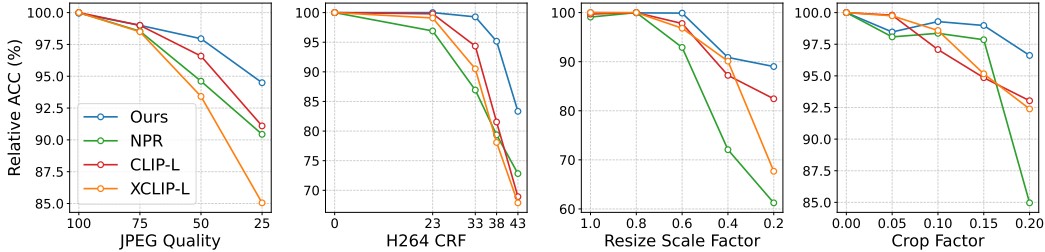

Figure 3: **Robustness on MovieGen under compression and spatial perturbations (relative ACC).** Perturbation methods include JPEG compression, H264 encoding, spatial resizing and cropping.

curacy peaking at 89.92 when using resolutions up to 720p. This confirms our hypothesis that maintaining aspect ratio and processing at higher resolutions are critical for capturing subtle, pixel-level forgery artifacts.

**Ablation Study on Temporal Resolution.** For all candidates, we sample the original videos at 2 fps and select random or center-aligned $T$ frames during training and testing, respectively. We observe that incorporating more temporal context is beneficial. Increasing the number of sampled frames ($T$) from 2 to 8 improves the average performance from 82.93 to 88.60. This suggests that longer sequences enhance the model's ability to detect temporal inconsistencies common in AI-generated videos.

**Ablation Study on Tuning method.** Regarding tuning strategies, full fine-tuning achieves the best average performance (88.60). Although the parameter-efficient LoRA approach significantly outperforms linear probing, full fine-tuning is justified for maximizing detection accuracy.

**Robustness Analysis**. We evaluate our model's robustness under common video perturbations, including compression, downscaling, and cropping, as shown in Figure 3. The model remains highly accurate under mild degradations such as moderate JPEG and H.264 compression. Performance drops become more pronounced with severe spatial changes. Notably, our model outperforms baselines under aggressive downscaling (scale $\leq 0.4$) and cropping (crop factor $\geq 0.15$), though all methods are affected by extreme spatial loss. These results highlight strong robustness to moderate noise and sensitivity to substantial spatial degradation.

## 5 CONCLUSION

In this work, we tackle two critical limitations in current AIGC detection methodologies: the reliance on outdated training data and the prevalent use of destructive fixed-resolution preprocessing. Our contributions are twofold. First, we introduce a comprehensive and up-to-date dataset comprising videos generated by 18 state-of-the-art generators, ensuring broader coverage of contemporary synthesis techniques. Second, we propose a novel detection framework capable of operating directly on videos at dynamic spatial-temporal resolutions. By leveraging the Qwen2.5-VL ViT backbone, our method avoids the information loss associated with downsampling, successfully preserving both fine-grained forgery artifacts and high-level semantic inconsistencies. Extensive evaluations demonstrate that our approach achieves state-of-the-art performance and substantially improves cross-generator robustness and generalization.

**Limitations.** Despite our efforts to construct a comprehensive dataset, the rapid evolution of generative models poses an ongoing challenge. Continuous data updates will be necessary to keep pace with emerging architectures. Additionally, processing videos at their native resolution inevitably incurs higher computational costs compared to downscaling-based methods, which may limit deployment in resource-constrained environments. Future work will focus on improving the computational efficiency of native-scale processing and investigating model explainability to deepen our understanding of the specific forensic traces leveraged for detection.

## 6 ACKNOWLEDGMENT

This work was supported by the project of Peng Cheng Laboratory (PCL2025A14), the National Natural Science Foundation of China (Grant No. 62350710797), and the Guangdong Basic and Applied Basic Research Foundation (Grant No. 2023B1515120065).

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

APPENDIX

This appendix provides a detailed analysis of our dataset, implementation details, additional experimental results, and visualizations:

- Section A: Data distribution and analysis of our dataset.
- Section B: Cross-validation experiment.
- Section C: Additional implementation details for both our method and the baseline methods.
- Section D: Additional experimental results and ablation studies.
- Section E: Visualizations and discussion.

## A    DATASET COMPOSITION

| Model / Video Source | Ver. | Availability | Videos | Resolution | FPS | Frame | Duration |
|---|---|---|---|---|---|---|---|
| **Real** Kinetics-400 (Kay et al., 2017) | 17.05 | real videos | 70K | 144p-720p | - | - | 5-10s |
| RepVideo (Si et al., 2025) | 25.01 | open-source | 4720 | 720x480 | 8 | 49 | 6.1s |
| Wan2.1 (Wan Team, 2025) | 25.01 | open-source | 4725 | 1280x720 | 16 | 81 | 5.0s |
| CausVid (5s) (Yin et al., 2025) | 25.01 | open-source | 4720 | 640x352 | 24 | 120 | 5.0s |
| Apple-STIV (Lin et al., 2024) | 24.12 | open-report | 4715 | 512x512 | 60 | 60 | 1.0s |
| Sora (Brooks et al., 2024) | 24.12 | private | 4720 | 854x480 | 30 | 150 | 5.0s |
| HunyuanVideo (Kong et al., 2024) | 24.12 | open-source | 4725 | 1280x720 | 24 | 129 | 5.4s |
| Gen-3 (Germanidis, 2024) | 24.06 | private | 4707 | 1280x768 | 24 | 256 | 10.7s |
| Luma (Lumalabs, 2024) | 24.06 | private | 4680 | 1360x752 | 24 | 121 | 5.0s |
| Kling (Kuaishou, 2024) | 24.06 | private | 4679 | 1280x720 | 30 | 153 | 5.1s |
| Jimeng (Jimeng AI, 2024) | 24.05 | private | 6214 | 1280x720 | 8 | 96 | 12.0s |
| OpenSora V1.1 (Zheng et al., 2024) | 24.04 | open-source | 4720 | 424x240 | 8 | 64 | 8.0s |
| Mira (Ju et al., 2024) | 24.04 | open-source | 4721 | 384x240 | 6 | 60 | 10.0s |
| VideoCrafter-2.0 (Chen et al., 2024a) | 24.01 | open-source | 4720 | 320x512 | 10 | 16 | 1.6s |
| Pika 1.0 (Pika Labs, 2023) | 23.11 | private | 4715 | 1280x720 | 24 | 72 | 3.0s |
| AnimateDiff-V2 (Guo et al., 2024) | 23.09 | open-source | 4715 | 512x512 | 8 | 16 | 2.0s |
| **Overall Fake** | - | - | 70,692 | 240-720p | 6-60 | 16-256 | 1-12s |

Table 8: **Statistics of real and synthetic videos in the proposed training set.**

| Model / Video | Split | Videos | Resolution | FPS | Frame | Duration |
|---|---|---|---|---|---|---|
| Movie Gen (Polyak et al., 2024) | validation (fake) | 1003 | 1920x1088 | 24 | 256 | 10.7s |
| Panda-70M (Chen et al., 2024d) | validation (real) | 1000 | 720p | 6-30 | - | 10-50s |
| Mixkit (mixkit, 2024) | test (real) | 215 | 720p | 15-60 | - | 10-17s |
| Pexels (pexels, 2024) | | 292 | 720p | 24-60 | - | 6-39s |
| Wan2.1 (Wan Team, 2025) | test (fake) | 215 | 1280x720 | 30 | 161 | 5.4s |
| Wan-1.3B (Wan Team, 2025) | | 292 | 832x480 | 16 | 81 | 5.0s |
| Hailuo (MiniMax, 2024) | | 215 | 1280x720 | 25 | 141 | 5.6s |
| Seaweed (Seawead et al., 2025) | | 215 | 1472x832 | 24 | 121 | 5.0s |
| Seedance (Gao et al., 2025) | | 215 | 1248x704 | 24 | 121 | 5.0s |
| StepVideo (Ma et al., 2025) | | 215 | 960×540 | 25 | 204 | 8.2s |

Table 9: **Statistics of real and synthetic videos in the proposed validation and Magic Videos Benchmark.**

### A.1    TRAINING SET.

Table 8 provides a comprehensive summary of the training dataset used in our work. Previous research has emphasized the critical importance of dataset quality and diversity in training robust detectors (Rajan et al., 2025), especially given the variety of artifacts produced by different generative models (Wu et al., 2025). To advance the field of AI-generated video detection, we curated a

large-scale dataset comprising outputs from 15 distinct video generation models. The majority of these synthetic videos are sourced from VBench (Huang et al., 2023), a benchmark selected for its high-quality prompt library and extensive evaluation of state-of-the-art models. This choice allowed us to avoid the costly and time-consuming processes of large-scale video filtering, quality control, and generation while ensuring high quality and consistency of generated video data.

Our dataset reflects the diverse and evolving landscape of video generation, featuring models developed between 2023 and 2025. It includes a wide range of model types in terms of availability (i.e., open-source, open-report, and private) and architecture (e.g., Diffusion U-Net, DiT-based, auto-regressive models, and others with undisclosed architectures). The models differ significantly in training methodology, data scale, output resolution, and video duration, contributing to a richly diverse training set.

To complement the synthetic videos, we sampled an equal number of real videos from Kinetics-400 (Kay et al., 2017). These were carefully selected to match the resolution, duration, and encoder distribution of the generated videos. This matching is essential for reducing potential biases and ensuring that the learned features are genuinely discriminative between real and fake content.

A key feature of our dataset is that all generative models were conditioned on the same prompt library, ensuring a shared semantic distribution across the generated videos. This unique setup enables controlled cross-validation experiments, allowing us to investigate inter-model relationships and identify key factors that influence detector performance, as discussed in Section B.

## A.2    VALIDATION AND TEST DATA

Table 9 presents the composition of our validation set and introduces a novel, high-quality Magic Videos Benchmark, which we name the Magic Videos Benchmark.

**Validation Set.**    Rather than adopting the common practice of partitioning a subset of the training data, we constructed the validation set from videos generated by Movie Gen (Polyak et al., 2024), a model that is architecturally and semantically similar but not identical to the models used in training. These synthetic videos are paired with 1,000 of real videos sampled from the Panda-70M (Chen et al., 2024d) dataset. During training, we apply early stopping based on the validation loss computed on this set. This strategy helps mitigate overfitting to the specific models and scenarios encountered during training, promoting the selection of a model checkpoint with stronger generalization capabilities.

**Magic Videos Benchmark.**    We identified a critical gap in existing benchmarks: they often lack coverage of the latest generative models and may exhibit evaluation biases. To address this, we constructed the Magic Videos Benchmark using a high-quality video generation pipeline, as introduced in Section 3 of the main paper. This benchmark includes real videos from two premium platforms—Mixkit(mixkit, 2024) and Pexels(pexels, 2024)—covering a diverse range of common scenes such as landscapes, architecture, human subjects, and news footage. These videos are provided at resolutions up to 1080p to ensure both high fidelity and content diversity. For evaluation, real and generated videos are matched into balanced subsets, allowing for the computation of accuracy and other performance metrics.

To generate the synthetic counterparts, we first applied ShareGPT4Video (Chen et al., 2024c) to produce high-quality captions for the real videos. These captions were then refined through a rigorous process of filtering, rewriting, and final prompt polishing. The resulting prompts were input to six advanced text-to-video models, comprising both open-source and commercial systems. Below, we detail the generative models used to construct the Magic Videos Benchmark:

- Wan2.1 (Wan Team, 2025): We used the Wanxiang platform API with the "professional" model, default settings, and prompt optimization disabled. Prompts were derived from the Mixkit collection. This model may apply post-processing, resulting in a higher frame rate than Wan-14B.

- Wan-1.3B (Wan Team, 2025): Videos were generated using the official open-source implementation and pre-trained model, with prompts from the Pexels collection.

- Hailuo (MiniMax, 2024): Accessed via the MiniMax-T2V-01 commercial API, this model was configured to generate 5-second videos using prompts from the Mixkit collection. Prompt optimization was not applied.

- Seaweed(Seawead et al., 2025): As official model weights are not publicly available, we used the commercial model Jimeng-S2.0(Jimeng AI, 2024), which is based on the Seaweed-alpha model. Prompts were sourced from the Mixkit collection. Generation was performed using prompts from the Mixkit collection.

- Seedance(Gao et al., 2025): In place of unavailable official weights, we used the commercial model Jimeng-S3.0(Jimeng AI, 2024), corresponding to the Seedance 1.0 Mini model. Prompts were sourced from the Mixkit collection.

- StepVideo (Ma et al., 2025): Videos were generated using the official API with the StepVideo-T2V endpoint (544px × 992px × 204f), using prompts from the Mixkit collection.

# B CROSS-VALIDATION EXPERIMENT

## B.1 EXPERIMENT SETUP

**Cross-Validation Setup.** This experiment focuses on in-domain, cross-model validation of detectors. The benchmark utilizes data generated by 15 models from VBench (Huang et al., 2023), which evaluates various generative models using a shared set of predefined prompts. Because all models generate videos from the same prompt library, we consider their outputs to belong to the same semantic domain. Let $F_i$ denote the subset of videos generated by model $i$, and let $R_0$ represent a fixed set of real videos, sampled to contain the same number of examples as each $F_i$. For each model $i$, we train a deepfake detector on the dataset $F_i, R_0$ and evaluate its performance on all other generated subsets $F_j$ (for $j \neq i$). This setup allows us to rigorously assess the generalization ability of detectors across different generative architectures while keeping the semantic domain fixed. It also provides a controlled environment for analyzing the relationships between generative model architectures and detection performance. This Cross-Validation Benchmark produces an $n \times n$ matrix $\mathbf{M}$, where $\mathbf{M}[i, j]$ represents the recall of a detection model trained on subset $i$ and evaluated on subset $j$. Based on preliminary observations, we propose the following two hypotheses, which will be validated in subsequent experiments.

**Similarity Between Generative Models.** The matrix entry $M[i, j]$ reflects the output similarity between generative models $i$ and $j$, influenced by factors such as model architecture, sampling strategies, and training data. We observe that models with more similar architectures tend to exhibit higher cross-validation accuracy between them. To quantify this relationship, we define a non-directional distance metric, $d(i, j) = 1 - 0.5 \times (M[i, j] + M[j, i])$. Using this metric, we apply Non-metric Multidimensional Scaling (MDS) (Kruskal, 1964) to produce a 2D spatial representation of the generative models. This visualization aids in understanding the architectural relationships and clustering patterns among the models, offering insights into how architectural similarity correlates with cross-detection performance.

**Impact of Generation Quality.** In addition to architecture, $M[i, j]$ is also influenced by the generation quality of model $i$. We hypothesize that higher-quality synthetic videos provide more realistic and informative supervision signals, enabling the classifier to learn more effective forgery-discriminative features. Since ground-truth quality labels are unavailable, we adopt scores from recent T2V benchmarks (Huang et al., 2023; Liu et al., 2024c; Huang et al., 2024) as a proxy for generation quality. To assess the relationship between generation quality and detection effectiveness, we compute Pearson correlation coefficients ($\rho$) between the benchmark quality scores and corresponding detection accuracies.

## B.2 CROSS-VALIDATION RESULTS

**Cross Validation.** As discussed above, we use the cross-validation matrix $\mathbf{M}$ to evaluate the similarity between generative models. Four detection models—F3Net(Qian et al., 2020), X-CLIP-B/32(Ni et al., 2022), TALL(Xu et al., 2023), and NPR(Tan et al., 2024)—are trained on 5K real videos from MSR-VTT and 5K generated videos from each specific model subset. These detectors

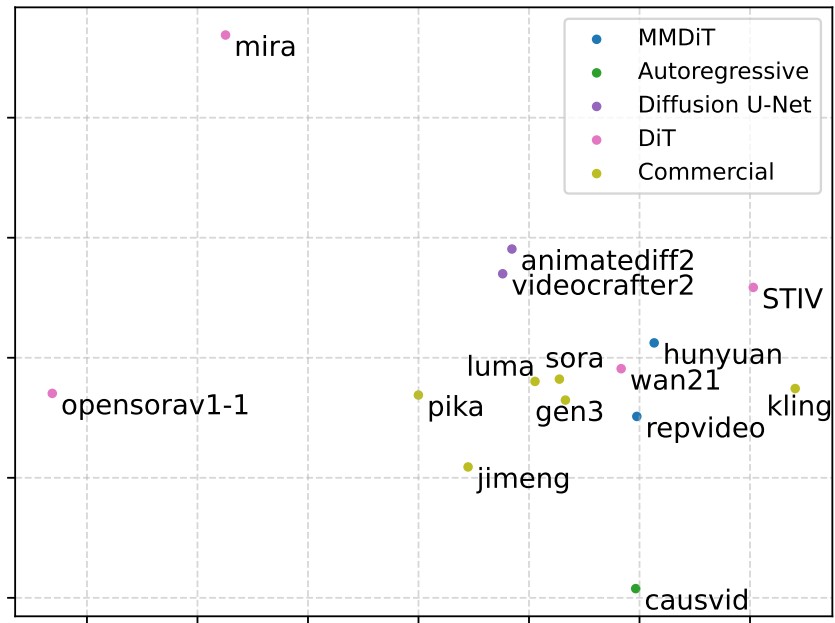

Figure 4: **MDS visualization of generator similarity induced by cross-model detection performance..** Model similarity is based on pairwise detection accuracy.

| Model | Wan21 | Hun-yuan | Kling | Sora | Gen-3 | Rep-Video | Jimeng | Luma | Mira | Pika | Open Sora | STIV | Caus Vid | VCraf ter-2 | ADiff -V2 | AVG |
|---|---|---|---|---|---|---|---|---|---|---|---|---|---|---|---|---|
| Wan21 | **99.7** | 97.4 | 93.1 | 97.1 | 98.4 | 88.7 | 96.8 | 97.5 | 33.8 | 91.3 | 44.7 | 88.3 | 97.7 | 92.1 | 94.2 | 87.38 |
| Hunyuan | 93.9 | **99.4** | 81.3 | 92.9 | 92.0 | 69.7 | 89.7 | 91.4 | 38.5 | 84.4 | 34.2 | 74.2 | 89.8 | 86.3 | 90.6 | 80.55 |
| Kling | 85.9 | 82.0 | **98.5** | 82.2 | 84.4 | 66.3 | 76.7 | 84.8 | 26.2 | 65.9 | 35.2 | 69.1 | 92.2 | 71.0 | 76.8 | 73.15 |
| Sora | 90.3 | 84.9 | 75.6 | **99.5** | 93.7 | 78.7 | 99.2 | 96.1 | 35.2 | 87.1 | 34.1 | 70.7 | 98.0 | 88.3 | 90.4 | 81.44 |
| Gen-3 | 92.6 | 84.9 | 81.7 | 93.0 | **99.6** | 83.8 | 94.4 | 95.3 | 23.7 | 88.1 | 50.6 | 75.1 | 96.5 | 80.4 | 84.5 | 81.60 |
| RepVideo | 97.0 | 91.5 | 92.8 | 96.9 | 98.3 | **98.8** | 98.6 | 97.8 | 29.4 | 94.8 | 56.6 | 86.9 | 99.8 | 91.7 | 91.1 | **88.13** |
| Jimeng* | 61.1 | 47.9 | 39.9 | 78.4 | 81.6 | 54.5 | **99.7** | 81.3 | 14.8 | 70.9 | 26.4 | 39.6 | 91.7 | 55.0 | 60.8 | 60.23 |
| Luma | 90.8 | 84.3 | 80.2 | 95.7 | 94.2 | 79.1 | 98.4 | **98.6** | 41.4 | 89.4 | 47.7 | 72.5 | 99.3 | 88.0 | 92.0 | 83.44 |
| Mira | 20.6 | 33.6 | 21.4 | 33.3 | 24.2 | 12.4 | 32.6 | 42.0 | **98.7** | 25.3 | 24.4 | 16.6 | 29.5 | 59.7 | 79.7 | 36.93 |
| Pika | 77.2 | 64.3 | 59.1 | 81.0 | 87.1 | 68.2 | 91.1 | 82.6 | 21.9 | **97.2** | 43.9 | 65.6 | 80.3 | 83.9 | 72.1 | 71.69 |
| Opensora V1.1 | 58.1 | 55.2 | 57.4 | 64.1 | 79.1 | 49.3 | 70.3 | 73.7 | 50.4 | 73.8 | **90.4** | 48.2 | 55.0 | 65.2 | 64.1 | 63.62 |
| Apple-STIV | 87.3 | 74.8 | 77.5 | 74.4 | 88.7 | 78.6 | 73.3 | 74.9 | 29.7 | 78.0 | 36.4 | **96.3** | 67.6 | 91.1 | 86.2 | 74.31 |
| CausVid | 28.7 | 21.8 | 19.6 | 39.6 | 34.5 | 38.3 | 50.7 | 44.8 | 7.3 | 16.0 | 7.4 | 16.7 | **99.6** | 18.4 | 28.8 | 31.47 |
| VideoCrafter-2 | 76.8 | 67.8 | 54.2 | 79.7 | 74.3 | 60.4 | 81.8 | 77.6 | 64.1 | 78.8 | 28.9 | 70.4 | 77.0 | **99.2** | 94.6 | 72.39 |
| AnimateDiff-V2 | 68.4 | 60.5 | 49.4 | 75.6 | 67.7 | 51.1 | 83.7 | 73.9 | 60.4 | 58.1 | 20.0 | 58.9 | 76.2 | 89.4 | **99.1** | 66.16 |

Table 10: **Cross-Validation Results**. Each cell in the table represents the average recall (%) of four detection models (NPR (Tan et al., 2024), TALL (Xu et al., 2023), X-CLIP-B/32 (Ni et al., 2022), F3Net (Qian et al., 2020)). The model is trained on generated videos of each subset and 5k real videos from MSR-VTT dataset.

are then tested on all other generative subsets. The average cross-validation accuracy across the four detectors is reported in Table 10. Each element in the table represents the mean detection accuracy across the four models. Diagonal entries correspond to in-subset evaluations, where the detector is tested on the same generative model used for training. As shown in Figure 4, we interpret the matrix **M** as a distance metric between generative models and apply Multidimensional Scaling (MDS) to project their relationships into a 2D space. This visualization reveals clusters of architecturally similar models, such as AnimateDiff2(Xu et al., 2024) and VideoCrafterV2(Chen et al., 2024a), while autoregressive-based models, such as (Yin et al., 2025), appear more distant from the rest. This mapping also informs a diverse training set selection of generative models, we could combine the cross validation accuracy and similarity to construct a high-quality and diverse dataset for data-efficient training.

**Better Generation, Better Detection.** In our cross-validation experiment, we observed that detection models trained on higher-quality generated videos exhibit stronger detection performance.

To validate this observation, we retrieved the overall VBench scores (Huang et al., 2023) for each generative model and conducted a correlation analysis between these scores and the average detection accuracies reported in Table 10. The results are visualized in Fig.1 of our main paper. Since the cross-validation data is directly sampled from VBench's evaluation set, the VBench scores provide an accurate proxy for the generation quality of each subset. Across 14 models (excluding CausVid, which features a fundamentally different model structure and training paradigm), we compute a Pearson correlation coefficient of $\rho = 0.86$ between average detection accuracy and VBench scores, indicating a strong positive correlation. Furthermore, when restricting the analysis to the six DiT-based models, the correlation increases to $\rho = 0.92$. These results strongly support our hypothesis: among models with similar architectures, higher-quality generation leads to better supervision signals, enabling detection models to learn more effective forgery-discriminative features.

## C    IMPLEMENTATION DETAILS

This section outlines the configurations and hyper-parameters used for training our proposed method, as well as the baseline models.

**Our Method.**    For our detector and Moon-ViT, all experiments are conducted using PyTorch with Automatic Mixed Precision (AMP) in bfloat16 to enable Flash Attention optimization and accelerate training. The visual backbone is initialized with Vision Transformer (ViT) weights from the officially released Qwen2.5-VL model. We explore multiple fine-tuning strategies with distinct hyperparameter settings: (1) Full fine-tuning: We set the batch size to 4 and train for 5 epochs with a learning rate of 1e-5. (2) Linear Probing (LP) and Parameter-Efficient Fine-Tuning (PEFT): These approaches use a larger batch size of 32 and a learning rate of 1e-4. Training continues for up to 30 epochs, with early stopping based on validation loss (patience = 5 epochs) to prevent overfitting.

**Other Baseline Methods.**    To ensure fair comparison, all baseline models are trained under a unified experimental setup. We used a consistent batch size of 32 and trained for a maximum of 30 epochs, also employing an early stopping strategy with a 5 epochs patience. The learning rate was adjusted based on the model architecture: for baselines utilizing a CLIP ViT backbone, such as X-CLIP and CLIP-based detectors, we set the learning rate to 1e-6; for all other models, a learning rate of 1e-5 was used.

**Data Pre-processing for Baseline Methods.**    A consistent data pre-processing pipeline is applied across all models during both training and testing. During training, each video is first sampled at a rate of 2 frames per second, from which 8 consecutive frames are extracted. If a video contains fewer than 8 frames, it is padded with blank frames to meet the required sequence length. Each frame is resized such that the shorter side is 224 pixels, followed by a random crop to a final resolution of 224×224. To enhance model robustness, we apply two forms of data augmentation: random horizontal flipping and random Gaussian noise. During testing, frames are sampled in the same manner as during training. After resizing the shorter side of each frame to 224 pixels, a center crop to 224×224 is applied instead of a random crop to ensure deterministic evaluation.

## D    ADDITIONAL RESULTS AND ABLATIONS

**Full Results on Genvideo-Val.**    As shown in Table 11, our proposed method achieves state-of-the-art performance across several key metrics. Notably, it attains an F1 score of 90.64 and an average precision (AP) of 96.13, surpassing all other leading methods—including DeMamba-CLIP, which was trained on the GenVideo dataset comprising 2.2 million samples. In contrast, our model was trained on only 140K samples, over ten times fewer, underscoring both the high quality of our training data and the efficiency of our method in learning robust forgery-discriminative features at native resolution. In addition, our model achieves a balanced accuracy (bACC) of 95.38, significantly outperforming all competing methods. This result demonstrates not only high overall detection performance but also the model's well-rounded and consistent capabilities across diverse forgery cases.

**Efficiency Comparison**.    As detailed in Table 12, we conduct a comprehensive efficiency analysis comparing our proposed Qwen2.5-VL ViT (Qwen2.5-ViT) with several strong baseline models.

| Model | Training Data | Metric | Sora | Morph Studio | Gen2 | HotShot | Lavie | Show-1 | Moon Valley | Crafter | Model Scope | Wild Scrape | Avg. |
|---|---|---|---|---|---|---|---|---|---|---|---|---|---|
| UNITE | FaceForensics++, SAIL-VOS-3D | Recall | 92.11 | 100.0 | 94.62 | 96.93 | 98.12 | 99.86 | 98.69 | 100.0 | 96.29 | 89.89 | 89.60 |
| | | F1 | - | | | | | | | | | | |
| | | AP | 88.57 | 100.0 | 100.0 | 90.16 | 89.91 | 98.34 | 99.52 | 100.0 | 98.96 | 92.56 | 92.76 |
| DeMamba-CLIP | GenVideo | Recall | 95.71 | 100.0 | 98.70 | 69.14 | 92.43 | 93.29 | 100.0 | 100.0 | 83.57 | 82.94 | **91.58** |
| | | F1 | 64.63 | 96.15 | 97.39 | 78.03 | 94.14 | 92.76 | 95.72 | 98.04 | 87.23 | 87.82 | 89.19 |
| | | AP | 85.50 | 100.0 | 99.59 | 76.15 | 96.78 | 96.99 | 99.97 | 100.0 | 89.80 | 89.72 | 93.45 |
| RINE | ProGAN | bACC | - | 84.00 | 89.10 | 66.00 | 96.70 | 91.80 | 85.70 | 98.30 | 76.60 | - | 74.10* |
| DeMamba | PyramidFlow | bACC | - | 83.80 | 92.20 | 62.00 | 79.60 | 72.60 | 92.40 | 87.50 | 68.60 | - | 78.10* |
| Corvi et al. | PyramidFlow | bACC | - | 97.00 | 98.80 | 81.40 | 95.50 | 92.10 | 98.40 | 98.30 | 97.10 | - | 94.30* |
| Ours | 15model-140k | Recall | 82.14 | 97.14 | 99.49 | 89.00 | 98.79 | 92.29 | 99.05 | 99.07 | 83.00 | 71.60 | 91.16 |
| | | F1 | 65.25 | 95.84 | 98.35 | 91.48 | 98.02 | 93.29 | 96.51 | 98.16 | 88.03 | 81.45 | **90.64** |
| | | AP | 82.49 | 99.36 | 99.95 | 96.55 | 99.78 | 97.88 | 99.87 | 99.89 | 94.50 | 90.98 | **96.13** |
| | | bACC | 90.87 | 98.38 | 99.55 | 94.31 | 99.20 | 95.95 | 99.33 | 99.34 | 91.31 | 85.61 | **95.38** |

Table 11: **Benchmarking Evaluation in terms of Recall, F1 score (F1), average precision (AP), and balance accuracy (bACC) on Genvideo-Val.** The results of RINE and DeMamba are reported in Corvi et al. (2025).

| Model | Resolution | #Params | FLOPS | Peak GPU Mem | Training Time / Epoch |
|---|---|---|---|---|---|
| CLIP-L | [224, 224] | 303.2M | 622.6G | 21.5GB (bs=4) 129.3GB (bs=32) | 9.5 A100 hours |
| X-CLIP-L | [224, 224] | 429.2M | 650.6G | 21.5GB (bs=4) 129.3GB (bs=32) | 10.5 A100 hours |
| Effort | [224, 224] | 0.2M/504.6M | 623.4G | 17.3G (bs=4) 75.1GB(bs=32) | 7.5 A100 hours |
| Qwen2.5-ViT | [224, 224] | 668.7M | 656G | 16.0GB(bs=4) | 2.3 A100 hours |
| Qwen2.5-ViT | dynamic [224p, 448p] | 668.7M | - | 37.9GB(bs=4) | 7 A100 hours |
| Qwen2.5-ViT-LoRA | dynamic [224p, 448p] | 2.6M/671.31M | - | 27.4GB(bs=4) | 5.5 A100 hours |

Table 12: Efficiency comparison results on model parameters, FLOPS, GPU memory utilization and time consumed during training.

For a standard input resolution of [224, 224], Qwen2.5-ViT exhibits remarkable training efficiency. Despite having more parameters (668.7M) than CLIP-L (303.2M), it achieves a $4.1\times$ reduction in training time (2.3 vs. 9.5 A100 hours) and a 25% decrease in peak GPU memory usage (16.0GB vs. 21.5GB at a batch size of 4). These gains are primarily attributed to efficiency-oriented design choices such as bfloat16 training and Flash Attention, which allow Qwen2.5-ViT to utilize computational resources more effectively. When adopting a dynamic resolution strategy, the training overhead naturally increases, yet Qwen2.5-ViT remains faster and more memory-efficient than the baselines. Moreover, our parameter-efficient fine-tuning variant, Qwen2.5-ViT-LoRA, requires updating only 2.6M parameters. This substantially reduces resource demands compared to full dynamic fine-tuning, lowering GPU memory from 37.9GB to 27.4GB and cutting training time from 7 to 5.5 hours. Overall, these results highlight that the superior efficiency of Qwen2.5-ViT stems from architectural optimizations, making the additional cost of higher dynamic resolutions acceptable in practice.

# E VISUALIZATION AND DISCUSSION

**Saliency Analysis**. We examine the model's attention responses to better understand its discriminative behavior, as illustrated in Figure 6. The results confirm that our native-resolution framework effectively captures two key types of features crucial for AIGC detection. (1) Low-level Artifacts: In billboard scenes, the model focuses on fine details such as distorted text rendering and unnatural edge transitions that are often lost during resolution downsampling. These high-frequency artifacts are indicative of generation errors and are critical for reliable detection. (2) High-level Semantics: In the fruit-cutting examples, the model attends to global inconsistencies, including object deformations and unrealistic lighting, suggesting it captures holistic content-level anomalies. This dual focus demonstrates that our approach leverages both spatial fidelity and semantic context, validating the design choice of preserving native resolution.

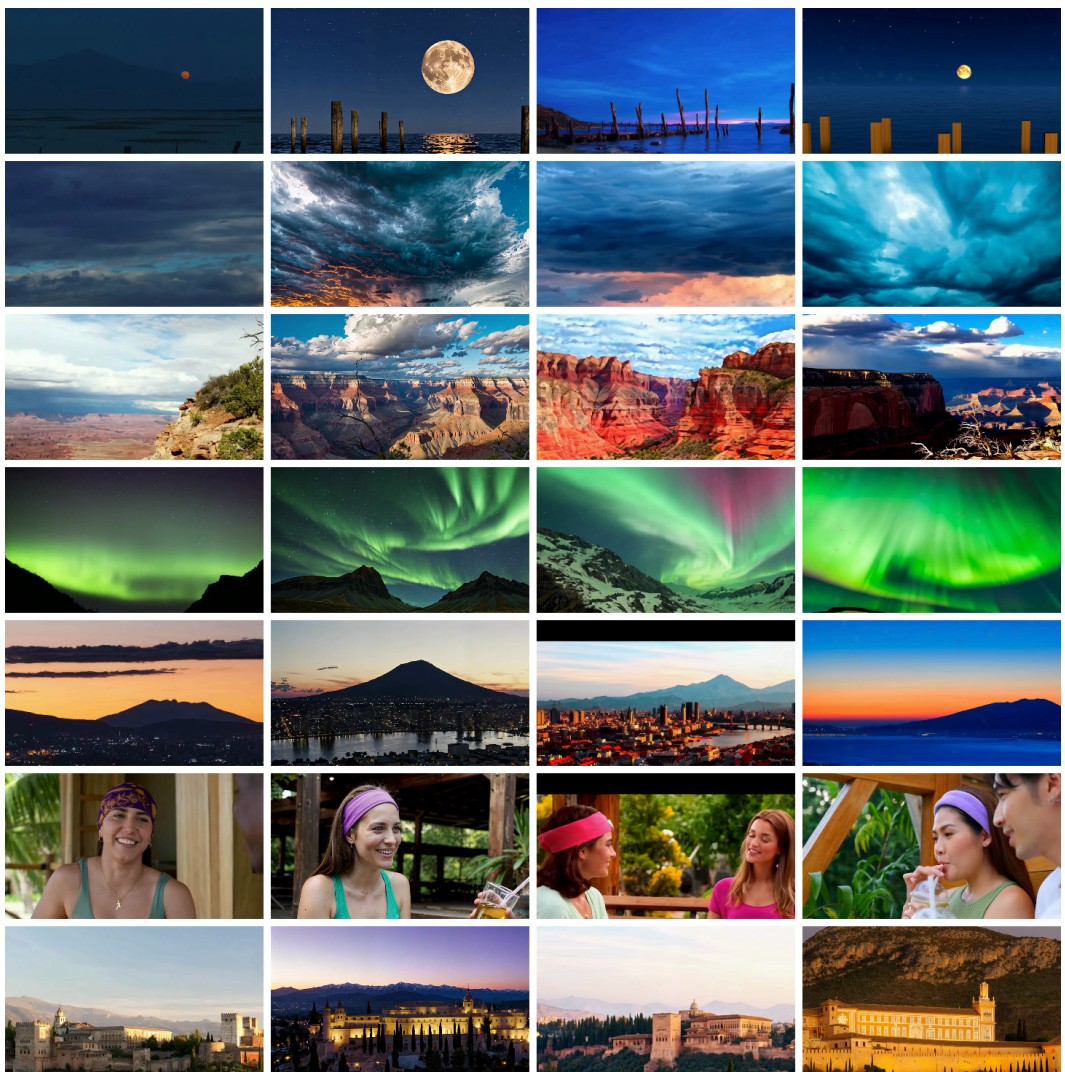

Figure 5: **Video Visualization from Magic Video Benchmark.** From left to right, each column denotes videos from real sources, seaweed, seedance, and wan2.1.

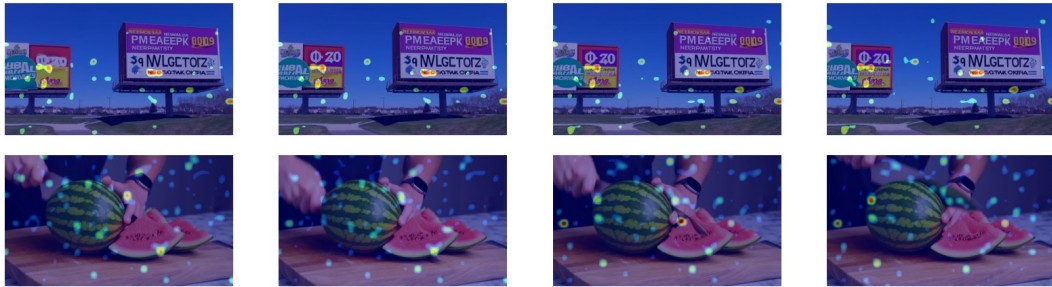

Figure 6: **Saliency Analysis.** Saliency maps of our model on AI-generated video samples.

Figures 5 and 7 to 9 present a selection of video samples from our dataset, with Figures 3–5 offering detailed visualizations along with their corresponding generative prompts. As illustrated in these figures, the videos in Magic Videos benchmark exhibit high visual quality, characterized by aesthetic appeal, rich motion, and diverse themes and visual effects.

By using carefully curated prompts to control the generative themes, we are able to evaluate a model's detection performance without introducing content bias. This methodological design promotes a fairer and more reliable assessment, encouraging the detector to learn generalizable forgery artifacts rather than memorizing specific object- or scene-level patterns.

**Acknowledgment of LLM Usage.** This manuscript has benefited from the assistance of a large language model, which was employed solely for grammar checking and language polishing. All scientific ideas, experimental designs, analyses, and conclusions are made by the authors.

*real video*

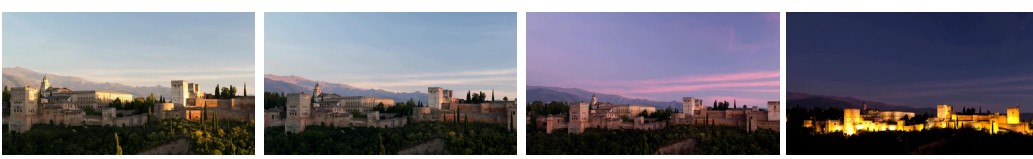

prompt: *The video showcases the Alhambra in Granada, Spain, transitioning from warm golden sunset tones to deep violet hues as night falls. The palatial structures, set against the Sierra Nevada mountains and lined with cypress trees, shift from sunlit brilliance to dramatic nighttime illumination. A subtle zoom enhances the view, while the changing light casts a striking contrast between the fortress's golden glow and the darkening sky, creating a captivating visual transformation.*

*generated video*

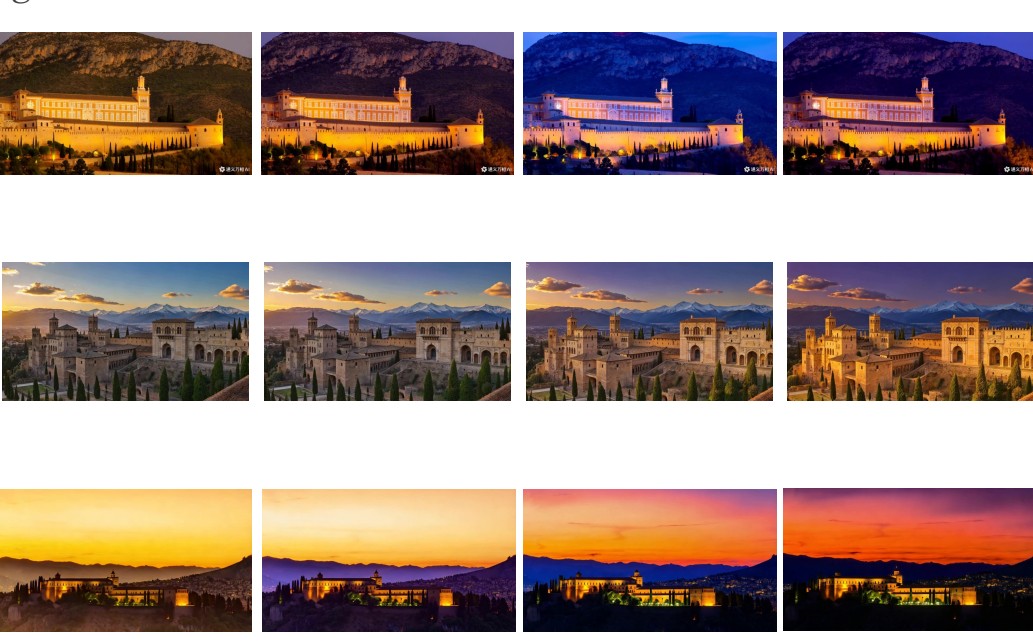

Figure 7: **Video Visualization from Magic Video Benchmark.**

*real video*

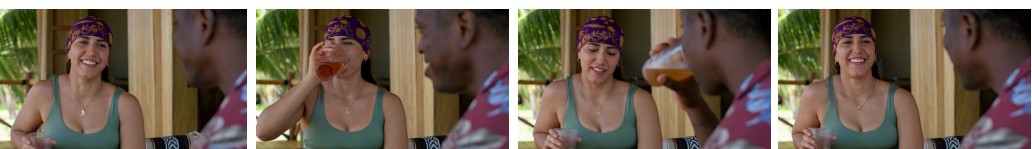

*prompt: A woman and a man engage in a friendly outdoor conversation amid wooden structures and greenery. The woman, wearing a purple headband and green tank top, sips her drink, signaling relaxation. Her expressions shift from savoring to engaging warmly, smiling and making eye contact. The man listens attentively, maintaining a steady demeanor. Both hold beverages, emphasizing the leisurely tone. Their uninterrupted dialogue features moments of humor and enjoyment in a serene setting.*

*generated video*

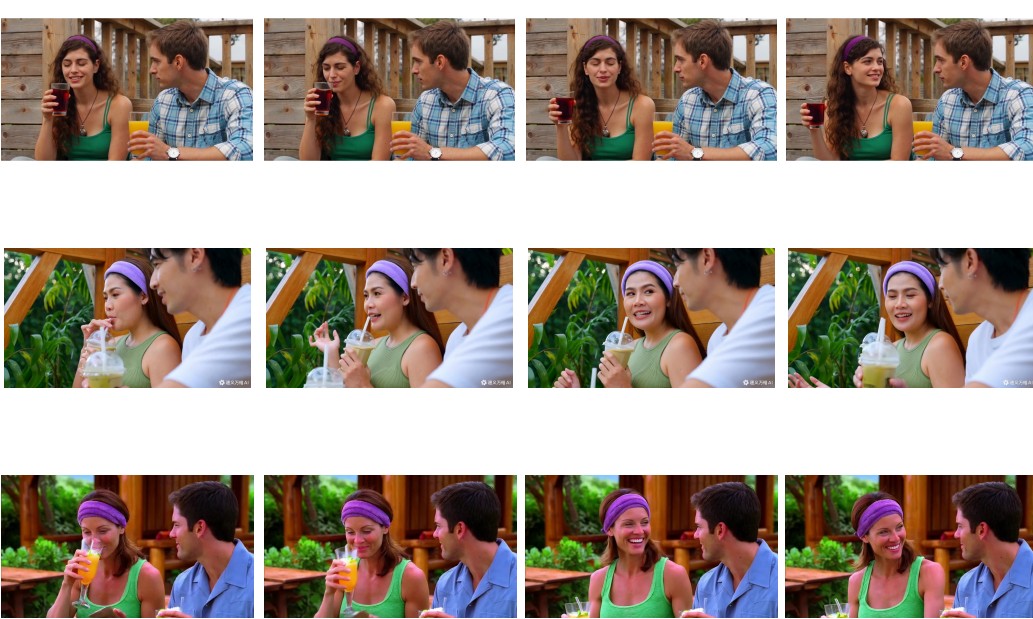

Figure 8: **Video Visualization from Magic Video Benchmark.**

*real video*

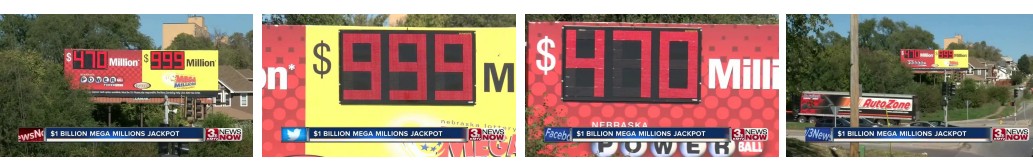

*prompt: The video showcases billboards for Powerball ($470M) and Mega Millions ($999M) under a sunny sky, with a '3 News Now' banner highlighting a '$1 BILLION MEGA MILLIONS JACKPOT.' Vibrant designs and mentions of 'NEBRASKA POWERBALL POWERPLAY' add local context. A brief error misstates the Mega Millions jackpot as $9M before correcting it. The video ends with a wide shot of the billboards against a residential backdrop, emphasizing their public appeal.*

*generated video*

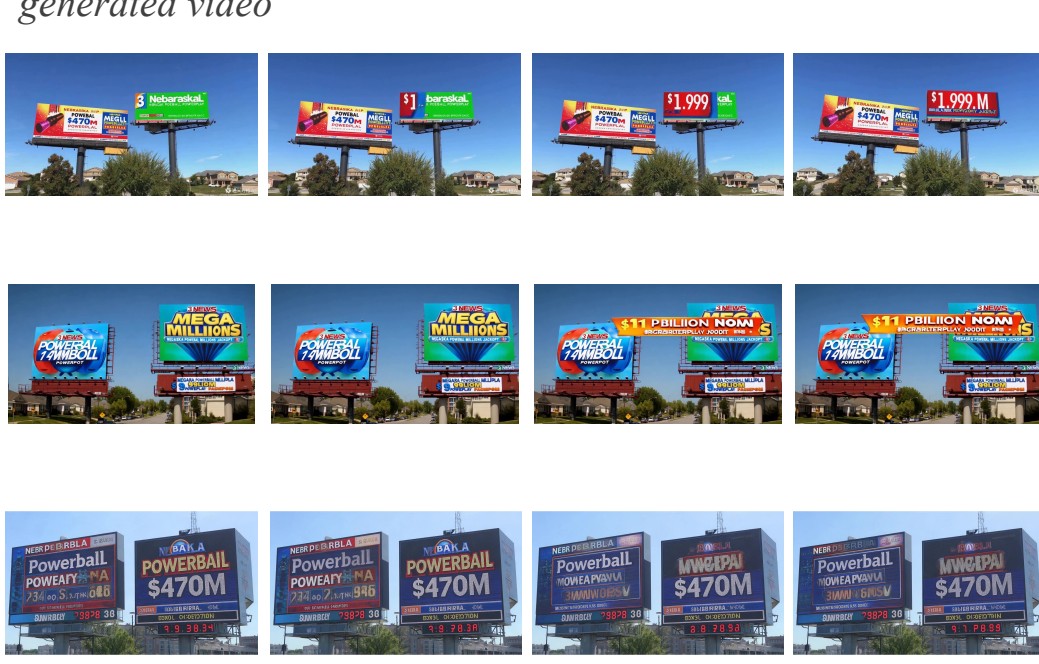

Figure 9: **Video Visualization from Magic Video Benchmark.**

