# OpenReview forum: "Preserving Forgery Artifacts: AI-Generated Video Detection at Native Scale"
_ICLR.cc/2026/Conference — ICLR 2026 Poster_

### Official Review · Reviewer_9HXd · 2025-10-27

**Soundness:** 2
**Presentation:** 3
**Contribution:** 3
**Rating:** 6
**Confidence:** 4

**Summary:**

This paper explores AI-generated video detection. It curates a dataset consisting of videos created from 15 or 18 generative models, as well as a novel framework that processes videos at their native spatial resolution and temporal duration, avoiding destructive preprocessing like resizing and cropping that mess up subtle forgery artifacts. The framework is built on the Qwen2.5-VL Vision Transformer, which uses 3D patchification directly on the raw video tensor, preserving the high-frequency details essential for detection. The experiment result shows improvement compared to baseline methods.

**Strengths:**

1. AI-generated video detection is an important topic
2. Providing a curated dataset of Gen-AI videos is a good contribution
3. The proposed detection framework shows improvement compared to baseline methods.

**Weaknesses:**

1. The proposed detection methodology is simply a combination of existing approaches and does not present a novel contribution

Detail: The core detection framework is a direct application of the existing Qwen2.5-VL Vision Transformer using its native 3D patchification strategy, which the authors note is adopted from prior work (Bai et al., 2025). While applying this to the forgery detection task is a valid contribution, the paper presents it as a 'novel detection framework'  when the architectural novelty itself is minimal. The primary novelty seems to be the dataset and the hypothesis that native-scale processing is superior, rather than a new detection architecture.

2. The presence of artifacts is vaguely assumed but not explicitly defined or proven to exist.

Detail: The authors do not define, visualize, or analyze what these artifacts are. A significant weakness is the lack of any model explainability (e.g., activation maps or gradient-based analysis) to prove that the native-scale model is actually focusing on these "pixel-level artifacts", while the 224p-resized model is not. The entire premise, while intuitive, is treated as an assumption rather than a proven scientific finding.

3. The use of downgraded input by cropping or resizing is argued to be the main fault of previous works. However, it is not explained or experimented with in connection with the actual baseline used.

Detail: Since the proposed method also performs poorly when given low resolution input, it is not surprising that baseline methods perform poorly when given low resolution input. This makes it impossible to know if the proposed method is superior due to its architecture or simply because it's the only one allowed to see the high-resolution data. A fair comparison would require adapting baselines to also accept high-resolution inputs or, at a minimum, comparing all models at the same fixed resolutions (e.g., 224p, 480p, 720p).

Other issues:
- The proposed detection method is shown to have a much higher cost in training with memory and GPU hours.
- Discrepancy in the number of generative models used (abstract: 16; Introduction: 15 and 18; appendix: 15)

**Questions:**

1. Can baseline detection methods be retrofitted to use the high-resolution inputs for training and inference?
2. Following the above question, if given high-resolution input in both training and inference, how does the baseline method compare with the proposed method?

3. Can the artifacts be identified or visualized?
4. Following the above question, can the artifacts be removed or interfered with so that the detection method fails?

---

> ### Author Response · Authors · 2025-11-22
>
> > **W1**: The proposed detection methodology is simply a combination of existing approaches and does not present a novel contribution.
>
> Thank you for the constructive feedback. We respectfully argue that our contribution lies not in designing a new network topology from scratch, but in identifying a critical bottleneck in current forgery detection pipelines and validating a **paradigm shift from "fixed-size preprocessing" to "native-resolution modeling."**
> We summarize our contribution as follows:
> 1. **Identifying the Bottleneck: Why Native Resolution Matters.** Existing detectors predominantly rely on fixed-size resizing or cropping. Through systematic cross-validation, we demonstrate that these standard preprocessing steps are the primary bottleneck for detecting high-quality AI-generated video.
>     - **Semantic Distortion:** Fixed resizing ignores the diverse aspect ratios of real-world content (unlike the 1:1 bias often found in generative models), introducing geometric distortions. Meanwhile, cropping leads to severe content loss and lack of global context.
>     - **Loss of Artifacts:** Down-sampling operations effectively act as low-pass filters, erasing the subtle, high-frequency artifacts that are crucial for identifying modern deepfakes.
>     - **Inefficiency:** While sliding windows can preserve resolution, they fail to model global dependencies and significantly increase inference costs.
> 2. The choice to build upon the Qwen2.5-VL architecture is a strategic decision to solve specific technical challenges that traditional backbones (e.g., ResNet, ViT) cannot handle efficiently
>     - **Variable Resolution Support:** Traditional models struggle with variable resolutions due to poor extrapolation of positional embeddings and the computational waste of padding within batches.
>     - **Advanced Modeling Capabilities:** Adapting older backbones to handle native resolutions and 3D temporal modeling would require fundamental re-engineering and massive-scale pre-training.
>     - By leveraging the advanced designs of Qwen2.5-VL (e.g., **RoPE** for extrapolation, **NaViT** + **Flash Attention** for efficiency, and **3D patchifying** for unified video modeling), we overcome the engineering barriers. This allows us to focus on the scientific contribution: proving that preserving the original spatial-temporal information is key to robust detection.
> 3. Beyond the method, we contribute a high-quality, multi-generator dataset and Magic Videos benchmark. We believe this work establishes a strong, reproducible baseline for the community. We commit to open-sourcing our dataset, code, and model weights to facilitate future research in AI-generated video detection.
>
> ---
> > **W2&Q3**: The presence of artifacts is vaguely assumed but not explicitly defined or proven to exist. Can the artifacts be identified or visualized?
>
> We agree that demonstrating what the model learns is crucial for validating our approach. We have conducted the requested attention visualization experiments and would upload the revised manuscript in a few days.
>
> Current effective AIGC detection typically relies on two complementary categories of discriminative features. The visualization results confirm that our native-resolution framework successfully captures both types of cues, justifying the necessity of our architectural choices:
>   - Capturing Low-level Artifacts: The attention maps show strong activation on fine-grained details, such as unnatural text rendering, inconsistent motion blur. Critically, these are the high-frequency features that are typically eroded by standard down-sampling
>   - Capturing High-level Semantics: The model also attends to global anomalies, such as unnatural object deformations (fruit-cutting videos) or distinct generative styles (e.g., "AI-like" lighting), proving it is not limited to local texture features.
>
> ---
>
> > **Q4**:  Following the above question, can the artifacts be removed or interfered with so that the detection method fails?
>
> The model tends to miss detections in scenarios that lack both high-frequency details and distinct semantic context, such as pure-sky scenes or static bird’s-eye landscapes. Moreover, when the input video is heavily compressed (e.g., with H.264 or JPEG compression), most forgery artifacts are suppressed. As a result, the model fails to identify any fake cues and tends to classify the video as real.
>
> Technically, if a generative model produces a video that is nearly pixel-identical to a real one, almost all detectors would fail. However, such samples are essentially harmless. Our goal is to exploit the limitations of generative models and identify the imperfections that arise when they attempt to synthesize content and details that does not exist in the real world.

---

> ### Author Response · Authors · 2025-11-22
>
> > **W3**: The use of downgraded input by cropping or resizing is argued to be the main fault of previous works. However, it is not explained or experimented with in connection with the actual baseline used. Detail: Since the proposed method also performs poorly when given low resolution input, it is not surprising that baseline methods perform poorly when given low resolution input. This makes it impossible to know if the proposed method is superior due to its architecture or simply because it's the only one allowed to see the high-resolution data. A fair comparison would require adapting baselines to also accept high-resolution inputs or, at a minimum, comparing all models at the same fixed resolutions (e.g., 224p, 480p, 720p).
> > **Q1**: Can baseline detection methods be retrofitted to use the high-resolution inputs for training and inference?
> > **Q2**: Following the above question, if given high-resolution input in both training and inference, how does the baseline method compare with the proposed method?
>
> We thank the reviewer for pointing out the necessity of a fair comparison regarding input resolution. We agree that distinguishing architectural superiority from the benefits of high-resolution input is crucial. To address this, we conducted additional experiments by retrofitting baseline methods to support high-resolution inputs and compared them directly with our proposed method.
> ### 1. **Experimental Design and Setup**
> Most baseline models (based on CNNs or ViTs) typically default to $224 \times 224$ resolution. To enable a fair comparison at higher resolutions ( $448 \times 448$ ):
>   - We retrained the baseline models (NPR, CLIP-L, and XCLIP-L) using both 224p and 448p inputs.
>   - For ViT-based models utilizing learnable absolute positional encodings (e.g., CLIP, XCLIP), we applied bicubic interpolation to extend the positional embeddings from $224 \times 224$ to $448 \times 448$ during the adaptation process.
>
> ### 2. **Experimental Results**
>
>   | | input resolution | Magic Videos (high-res) | Genvideo (low-res) | AVG |
>   |---|---|---|---|---|
>   | NPR | 224p | 67.09 | 86.75 | 76.92 |
>   | NPR | 448p | 70.15 | 84.97 | 77.56 |
>   | CLIP-L | 224p | 72.51 | 94.00 | 83.26 |
>   | CLIP-L | 448p | 74.60 | 91.93 | 83.27 |
>   | XCLIP-L/14 | 224p | 70.22 | 94.74 | 82.48 |
>   | XCLIP-L/14 | 448p | 75.14 | 92.43 | 83.79 |
>   | Qwen2.5-ViT (Ours) | dynamic [224p, 448p] | **77.01** | **96.01** | **86.51** |
>
> **Observations:**
>   - **Trade-off in Baselines:** As hypothesized, increasing the input resolution to 448p for baselines yields clear improvements on the high-resolution *Magic Videos* dataset (e.g., CLIP-L improves from 72.51% to 74.60%). However, this comes at a cost: performance on the lower-resolution *Genvideo* dataset degrades (e.g., CLIP-L drops from 94.00% to 91.93%). This suggests that fixed high-resolution resizing introduces up-scale artifacts when processing low-resolution content.
>   - **Superiority of Our Method:** Our model, which utilizes dynamic resolution (handling inputs at their native scale between 224p and 448p), achieves the best performance across both datasets (77.01% and 96.01%).
>
> ### 3. **Analysis and Discussion**
>
> The results highlight the architectural limitations of earlier backbones compared to modern designs:
> - **Positional Encodings:** Traditional baselines often rely on absolute positional encodings trained at a specific scale. While interpolation allows them to accept larger inputs, it does not fundamentally solve the issue of resolution adaptation, leading to the observed "see-saw" performance trade-off.
> - **Efficiency and Robustness:** Simply scaling up baselines significantly increases computational overhead without guaranteeing consistent gains across diverse datasets.
> - **Advantage of Modern Architecture:** Our method adopts a more modern visual encoding strategy tailored for variable resolutions. By leveraging Rotary Positional Embeddings (RoPE) combined with NaViT processing pipeline and Flash Attention, our model balances the additional computational cost of high-resolution features with the flexibility to handle low-resolution inputs natively. This design allows the model to extract fine-grained details when available without compromising robustness on lower-quality data, making it significantly more suitable for the generated video detection task.

---

### Official Review · Reviewer_KYH3 · 2025-10-31

**Soundness:** 3
**Presentation:** 3
**Contribution:** 2
**Rating:** 4
**Confidence:** 5

**Summary:**

The paper proposes an AI-generated video detector that (1) builds a  dataset and (2) trains a detector on native spatial resolution and temporal duration using Qwen2.5-VL ViT with 3D patchification, so that high-frequency, position-dependent artifacts aren’t destroyed. On three families of benchmarks (GenVideo, DVF, and their Magic Videos), it reports consistently higher performance than prior image-detectors, deepfake-detectors, and video backbones. The core narrative is: current detectors are undertrained on modern, high-quality video generators, and downsampling design.

**Strengths:**

1. Timely problem & data refresh. Most detectors are indeed lagging behind 2024–2025 video generators; this paper explicitly targets that gap and gathers content from new models, including commercial/API ones. That’s rare and useful.
2. Clear empirical story about resolution. The experiment results supports the authors' claim well and the motivation is clear.
3. The paper is well-written and easy to follow.

**Weaknesses:**

1. The paper addresses two real but relatively well-recognized issues — resolution-destructive preprocessing and stale generator distributions. The proposed solution mainly combines (i) a recent, high-capacity video/VL backbone and (ii) a refreshed multi-generator dataset. The detector architecture itself is largely standard, without forgery-specific inductive biases. Data curation process is also well defined in previous fake video detection work. Thus the contribution is more of an engineering consolidation.
2. Dataset release / licensing / reproducibility is unclear. Several of the listed video sources (Kuaishou, Luma, MovieGen) are API/commercial. The paper doesn’t yet make it clear what exactly will be released, how prompts will be shared, and how others can reproduce “Magic Videos”. For ICLR this is important.
3. No fairness breakdown. Real videos come from different sources than synthetic ones; there can be source, watermark, or codec biases. The paper doesn’t fully rule that out. It's important to demonstrate the performance on unseen real videos from very different source such as KITTI, etc. to demonstrate the generalizability of the proposed model.
4. Limited analysis on real-world perturbations. The core claim is about preprocessing destroying artifacts, but actual attackers / platforms will introduce their own compress-and-resize chains. It would be good to see: scale jitter & heavy H.264/HEVC compression and see whether “native-scale” still wins.
5. Incomplete discussion of most recent related work such as [1, 2].

Reference
1. Distinguish Any Fake Videos: Unleashing the Power of Large-scale Data and Motion Features. 2024
2. How Far are AI-generated Videos from Simulating the 3D Visual World: A Learned 3D Evaluation Approach. 2025

**Questions:**

1. How sensitive is the detector to platform compression? Your main argument is “don’t downsample to 224.” But if a platform already did that, can your model still outperform the older 224-trained models? A controlled experiment with platform-style compression would be convincing.
2. How do you prevent source leakage? Since real videos come from specific real video datasets and generated ones are from VBench/MovieGen/etc., detectors might be learning source signatures. Do you have a cross-source test where the real videos share encoding with the synthetic ones?
3. Happy to see more discussion with latest related work in the related area. This would help make the contribution clear.

---

> ### Author Response · Authors · 2025-11-22
>
> > **W1**: The paper addresses two real but relatively well-recognized issues — resolution-destructive preprocessing and stale generator distributions. The proposed solution mainly combines (i) a recent, high-capacity video/VL backbone and (ii) a refreshed multi-generator dataset. The detector architecture itself is largely standard, without forgery-specific inductive biases. Data curation process is also well defined in previous fake video detection work. Thus the contribution is more of an engineering consolidation.
>
> Thank you for the constructive feedback. We respectfully argue that our contribution lies not in designing a new network topology from scratch, but in identifying a critical bottleneck in current forgery detection pipelines and validating a **paradigm shift from "fixed-size preprocessing" to "native-resolution modeling."**
> We summarize our contribution as follows:
> 1. **Identifying the Bottleneck: Why Native Resolution Matters.** Existing detectors predominantly rely on fixed-size resizing or cropping. Through systematic cross-validation, we demonstrate that these standard preprocessing steps are the primary bottleneck for detecting high-quality AI-generated video.
>     - **Semantic Distortion:** Fixed resizing ignores the diverse aspect ratios of real-world content (unlike the 1:1 bias often found in generative models), introducing geometric distortions. Meanwhile, cropping leads to severe content loss and lack of global context.
>     - **Loss of Artifacts:** Down-sampling operations effectively act as low-pass filters, erasing the subtle, high-frequency artifacts that are crucial for identifying modern deepfakes.
>     - **Inefficiency:** While sliding windows can preserve resolution, they fail to model global dependencies and significantly increase inference costs.
> 2. The choice to build upon the Qwen2.5-VL architecture is a strategic decision to solve specific technical challenges that traditional backbones (e.g., ResNet, ViT) cannot handle efficiently
>     - **Variable Resolution Support:** Traditional models struggle with variable resolutions due to poor extrapolation of positional embeddings and the computational waste of padding within batches.
>     - **Advanced Modeling Capabilities:** Adapting older backbones to handle native resolutions and 3D temporal modeling would require fundamental re-engineering and massive-scale pre-training.
>     - By leveraging the advanced designs of Qwen2.5-VL (e.g., **RoPE** for extrapolation, **NaViT** + **Flash Attention** for efficiency, and **3D patchifying** for unified video modeling), we overcome the engineering barriers. This allows us to focus on the scientific contribution: proving that preserving the original spatial-temporal information is key to robust detection.
> 3. Beyond the method, we contribute a high-quality, multi-generator dataset and Magic Videos benchmark. We believe this work establishes a strong, reproducible baseline for the community. We commit to open-sourcing our dataset, code, and model weights to facilitate future research in AI-generated video detection.
> ---
> > **W2**: Dataset release / licensing / reproducibility is unclear. Several of the listed video sources (Kuaishou, Luma, MovieGen) are API/commercial. The paper doesn’t yet make it clear what exactly will be released, how prompts will be shared, and how others can reproduce “Magic Videos”. For ICLR this is important.
>
> The generated videos from Magic Videos will be released and made freely available for academic research. In addition, we will also share the prompts to support reproducibility and further exploration.
>
> ---
> > **W5&Q3**: Incomplete discussion of most recent related work such as [1, 2].
> > [1] Distinguish Any Fake Videos: Unleashing the Power of Large-scale Data and Motion Features. 2024
> > [2] How Far are AI-generated Videos from Simulating the 3D Visual World: A Learned 3D Evaluation Approach. 2025
>
> We appreciate your valuable advice, and we will incorporate the relevant literature review and potential experiments into the revised manuscript.
>
> Paper [1] introduces GenVidDet, a comprehensive large-scale dataset for AI-generated video detection, and proposes the Dual-Branch 3D Transformer (DuB3D), which robustly distinguishes real from fake content by effectively fusing spatial-temporal features with motion cues derived from optical flow. Paper [2] introduces Learned 3D Evaluation (L3DE), a novel framework that leverages a 3D convolutional network trained on monocular cues of appearance, motion, and geometry to quantitatively and interpretably assess the 3D visual coherence of AI-generated videos without relying on explicit 3D reconstruction.
>
> Both papers are closely related to our work and contribute meaningful insights to the field of AIGC detection.

---

> ### Author Response · Authors · 2025-11-22
>
> > **W3&Q2**: No fairness breakdown. Real videos come from different sources than synthetic ones; there can be source, watermark, or codec biases. The paper doesn’t fully rule that out. It's important to demonstrate the performance on unseen real videos from very different source such as KITTI, etc. to demonstrate the generalizability of the proposed model. How do you prevent source leakage? Since real videos come from specific real video datasets and generated ones are from VBench/MovieGen/etc., detectors might be learning source signatures. Do you have a cross-source test where the real videos share encoding with the synthetic ones?
>
> We thank the reviewer for this critical question. We fully agree that preventing source leakage (shortcuts from codecs, resolutions, or specific data sources) is vital for a robust forensic detector.
>
> To strictly rule out these confounding factors, we implemented a multi-stage strategy covering dataset construction, preprocessing, and extensive cross-domain evaluation.
>
> ### 1. Mitigation of Source Leakage via Construction & Preprocessing
> - **Statistical Alignment:** During dataset construction, we carefully balanced the real and synthetic subsets to ensure no significant statistical bias existed in terms of semantic topic, video duration, or native resolution.
>
> - **Unified Preprocessing Pipeline**: We applied a unified re-encoding protocol to all data. Both real videos and generated videos in the training set were encoded using H.264 codec and preprocessed with our dynamic resolution pipeline. This step effectively removes specific codec signatures, quantization tables, or metadata artifacts that could serve as "shortcuts," ensuring the model focuses solely on visual content.
>
> - **Cross-domain evaluation:** three different test sets reported in our paper differ greatly from the training set. These evaluations are all cross-domain, cross-source, and cross-generator. The results demonstrate, to some extent, the generalization ability of both our dataset and our model.
>
> ### 2. Evaluation on Unseen Real-World Domains
>
> We follow the suggestion that testing on distinct real sources to verify generalization and evaluate our model on real videos sampled from four additional external real-video datasets covering diverse scenarios: **BDD100K** (autonomous driving), **TGIF** (low-resolution GIFs), **Ego4D** (egocentric), and **InternVid-10M** (YouTube).
>
>   | **Dataset** | **Scenario** | **Num Samples** | **Qwen2.5-ViT (Ours) ACC** |
>   |---|---|---|---|
>   | **BDD100K** | Driving / Dashcam | 1K | 87.71% |
>   | **TGIF** | Low-Res GIF | 10K | 94.11% |
>   | **Ego4D** | Egocentric / Wearable | 500 | 89.33% |
>   | **InternVid-10M** | General / YouTube | 2K | 93.89% |
>
> The model maintains high accuracy (>87-94%) across these unseen domains. Notably, the robustness on *TGIF* (noisy, low-res) and *BDD100K* (specific driving scenes) confirms that our model has learned generalizable features of "reality" rather than overfitting to the specific source characteristics of our training set.
>
> ### 3. Evaluation on DeepTraceReward Benchmark
> To further demonstrate robustness against unseen generators, we evaluated our method on the *DeepTraceReward* benchmark [1], which contains 4,335 videos from 7 recent models (including Pika-1.5, Kling-1.5, etc). We compared our Qwen2.5-ViT against leading Multimodal LLMs reported in [1].
>
>   | Method | ACC | Fake ACC | Real ACC |
>   |---|---|---|---|
>   | GPT-5 | 90.7 | 84.6 | 98.8 |
>   | GPT-4.1 | 92.9 | 89.1 | 97.9 |
>   | Gemini 2.5 Pro | 84.3 | 75.7 | 95.8 |
>   | Qwen 2.5 VL 7B | 51.7 | 20.2 | 93.4 |
>   | DeepTraceReward (w/ Qwen2.5 VL 7B) | 74.7 | 55.7 | 100.0 |
>   | Qwen2.5-ViT (Ours) | 97.2 | 96.3 | 98.2 |
>
> - **Superior Generalization:** Our model achieves **97.2% accuracy**, significantly outperforming massive foundation models (e.g., GPT-5, Gemini 2.5 Pro) on the binary classification task.
> - **Balanced Detection:** While general-purpose VLMs often struggle with detecting fakes (showing lower Fake ACC), our model demonstrates balanced performance (96.3% Fake ACC vs. 98.2% Real ACC), proving its effectiveness in identifying artifacts from the latest generation engines without overfitting to specific training generators.
>
> [1] Reference: "Learning Human-Perceived Fakeness in AI-Generated Videos via Multimodal LLMs"

---

> ### Author Response · Authors · 2025-11-22
>
> > **W4** Limited analysis on real-world perturbations. The core claim is about preprocessing destroying artifacts, but actual attackers / platforms will introduce their own compress-and-resize chains. It would be good to see: scale jitter & heavy H.264/HEVC compression and see whether “native-scale” still wins.
> > **Q1** How sensitive is the detector to platform compression? Your main argument is “don’t downsample to 224.” But if a platform already did that, can your model still outperform the older 224-trained models? A controlled experiment with platform-style compression would be convincing.
>
> We agree that robustness against real-world degradation chains is essential for deployment. We have addressed this through existing analysis and new controlled experiments on video compression.
> 1. **Image-Level Perturbations**: In **Supplementary Section D and Figure 4**, we have presented the robustness of our model against image-level perturbations, including JPEG compression, gaussian noise, random cropping, and resizing on *Magic Videos*. The results demonstrate strong resilience. To facilitate a direct comparison, we will add the performance curves of baseline models under these perturbations to the revised version.
> 2. **Impact of Heavy Video Compression (H.264)**: We conduct a robustness test on the *MovieGen* dataset. We applied secondary encoding using the standard libx264 codec with varying Constant Rate Factors (CRF) ranging from 23 (high quality) to 43 (heavy compression/degradation).
>     - | ACC (%) on MovieGen-2k | original | CRF=23 | CRF=33 | CRF=38 | CRF=43 |
>   |---|---|---|---|---|---|
>   | NPR | 87.2 | 84.5 | 75.8 | 69.2 | 63.5 |
>   | CLIP-L | 99.1 | 98.9 | 93.5 | 80.8 | 68.3 |
>   | XCLIP-L | 99.0 | 98.1 | 89.6 | 77.3 | 67.2 |
>   | Qwen2.5-ViT (Ours) | 97.2 | 97.2 | 96.5 | 92.5 | 81.0 |
>     - **Sensitivity of SOTA Baselines:** While state-of-the-art baselines like **CLIP-L** and **XCLIP-L** perform excellently on high-quality data (~99%), they suffer catastrophic degradation under heavy compression. At CRF 43, their accuracy plummets to **68.3%** and **67.2%**, respectively, dropping by over 30 points. This confirms that these 224-trained models rely heavily on high-frequency details that are lost during platform-style compression.
>     - **Robustness of Our Method:** Under the same severe compression (CRF 43), our model maintains an accuracy of **81.0%**, significantly outperforming NPR-224p (63.5%), CLIP-L (68.3%), and XCLIP-L (67.2%).
> 3. **Robustness Across Different Codecs**
> 	  To verify that our model is not biased towards a specific encoder, we evaluated performance on *MovieGen-2K* re-encoded with different codecs (H.264, H.265, MPEG4) at comparable high-quality settings.
>     - | **Codec on MovieGen-2k** | **ACC (%)** | **AP (%)** |
>   |---|---|---|
>   | Original | 97.2 | 99.0 |
>   | libx264 (CRF=23) | 97.2 | 98.9 |
>   | libx265 (CRF=28) | 97.3 | 98.9 |
>   | mpeg4 (qv=3) | 97.2 | 99.0 |
>
>     - **Result:** The performance variance across different codecs is negligible (<0.1%). This confirms that our detector is identifying intrinsic generation anomalies rather than overfitting to specific compression signatures or codec biases.

---

### Official Review · Reviewer_X5mz · 2025-10-31

**Soundness:** 3
**Presentation:** 2
**Contribution:** 3
**Rating:** 6
**Confidence:** 3

**Summary:**

This paper investigates the limitations of AI-generated video detection, highlighting that fixed-resolution preprocessing removes high-frequency forgery artifacts and that outdated datasets struggle to represent modern models. The authors build a dataset of about 140K videos from 18 generative models and real sources such as Kinetics and MSVD, and design a realism-oriented test set called Magic Videos. They propose a detection framework based on Qwen2.5-VL ViT, which processes videos at their native spatial and temporal scales using 3D patch tokenization to preserve forgery details. Experiments are conducted on GenVideo, DVF, and Magic Videos.

**Strengths:**

Builds a large-scale dataset with over 140,000 videos from 18 generative models, covering mainstream AI video generation technologies and offering a research resource.

Conducts experiments on multiple benchmarks, including the self-built Magic Videos test set and public datasets, ensuring comparability and reliability.

Employs a dynamic resolution processing framework that preserves detection performance and demonstrates generalization ability across datasets.

Designs ablation experiments that are systematic and support the main conclusions.

Integrates Flash Attention and LoRA optimizations, reflecting concern for efficiency and practical deployment.

**Weaknesses:**

Lacks analysis of the model’s decision-making mechanism, failing to verify whether the model truly learns forgery-related cues rather than biased features in the data.

Provides insufficient discussion of computational efficiency, with no concrete results on inference speed or resource requirements for real-world deployment.

The training data source is limited, relying mainly on the VBench dataset, which may introduce hidden biases.

The failure case analysis is inadequate, lacking exploration of the scenarios and causes where the model fails.

**Questions:**

Regarding computational efficiency, the paper mentions optimization techniques such as Flash Attention, but lacks specific key metrics such as inference speed and memory usage. Could you provide detailed efficiency data under typical hardware configurations to assess the feasibility of this method in real-world scenarios?

For Magic Videos, how is the "indistinguishable to humans" claim verified?

Experimental results show significant performance differences across different generators (from 72.26% to 85.12% in Table 2). Have you analyzed the specific reasons for these differences? Are there any specific types of generated content or technical approaches that are blind spots for the current method?

Could you provide a more in-depth interpretability analysis, such as attention visualization or feature analysis, to demonstrate that the model truly learns meaningful forgery traces, rather than relying on other superficial features in the data?

---

> ### Author Response · Authors · 2025-11-22
>
> > **W1&Q4**: Lacks analysis of the model’s decision-making mechanism, failing to verify whether the model truly learns forgery-related cues rather than biased features in the data.
>
> Thank you for your constructive feedback. We agree that demonstrating what the model learns is crucial for validating our approach. We have conducted the requested attention visualization experiments and would upload the revised manuscript in a few days.
>
> Current effective AIGC detection typically relies on two complementary categories of discriminative features. The visualization results confirm that our native-resolution framework successfully captures both types of cues, justifying the necessity of our architectural choices:
>   - Capturing Low-level Artifacts: The attention maps show strong activation on fine-grained details, such as unnatural text rendering, inconsistent motion blur. Critically, these are the high-frequency features that are typically eroded by standard down-sampling
>   - Capturing High-level Semantics: The model also attends to global anomalies, such as unnatural object deformations (fruit-cutting videos) or distinct generative styles (e.g., "AI-like" lighting), proving it is not limited to local texture features.
>
> Failure Case Analysis. To further validate this, we analyzed failure cases. The model tends to miss detections in scenarios lacking both high-frequency details and distinct semantic context, such as pure blue sky. This paradoxically confirms that the model is indeed searching for specific forensic traces (semantic or textural) rather than overfitting to superficial dataset biases or shortcut learning.
>
> ---
>
> > **W2&Q1:** Provides insufficient discussion of computational efficiency, with no concrete results on inference speed or resource requirements for real-world deployment.
>
> Thank you for raising the concern about computational efficiency. We acknowledge that processing high-resolution inputs typically implies higher costs. However, we would like to clarify that our proposed method achieves superior efficiency compared to baselines due to its systematic architectural optimizations, specifically the dynamic resolution mechanism and Flash Attention integration.
>
> 1. **Training Efficiency (Referencing Appendix Table 11):** As demonstrated in our Appendix (Page23, Table11), although our backbone may have higher parameters or theoretical FLOPs, the actual training cost is lower.
> |Model|Input Resolution|#Params|FLOPs|Peak GPU Mem|Training Time / A100 hrs per epoch|
> |--|--|--|--|--|--|
> |CLIP-L|224 × 224|303.2M|622.6G|129.3GB|9.5h|
> |XCLIP-L|224 × 224|429.2M|650.6G|129.3GB|10.5h|
> |Effort|224 × 224|0.2M/504.6M|623.4G|75.1GB|7.5h|
> |Qwen2.5-ViT (Ours)|224 × 224|668.7M|656G|16.0GB|2.3h|
> |Qwen2.5-ViT (Ours)|dynamic [224p, 448p]|668.7M|656G|37.9GB|7h|
> |Qwen2.5-ViT-Lora (Ours)|dynamic [224p, 448p]|2.6M / 671.3M|656G|27.4GB|5.5h|
> 2. **Inference Speed Comparison (New Experiment):** To further address the concern regarding real-world deployment, we conducted a detailed inference speed comparison (measured on a single A100 GPU). The results are summarized in the table below:
> ||input resolution|frame|inference time|fps|
> |--|--|--|--|--|
> ||||||
> |CLIP-L|224p|1|3m20s|90.9|
> |CLIP-L|224p|8|24m20s|99.6|
> |XCLIP-L|224p|8|26m34s|93.3|
> |Effort|224p|8|25m1s|96.9|
> |Qwen2.5-ViT (Ours)|force 224p|8|5m27s|444.9|
> |Qwen2.5-ViT (Ours)|dynamic [224p, 448p]|8|13m59s|177.2|
>
>   - **Experiment Settings**: All methods are evaluated on GenVideo-val, which contains 19k videos in resolution from 224p to 1280p. For a fair comparison, we use temporal padding for our method to ensure 8-frame input of all videos.
>   - **Observation:** Our method (using 8 frames with dynamic high resolution) is significantly faster than baseline methods with 8-frame input.
>   - **Key Finding:** Surprisingly, our 8-frame inference speed is comparable to the single-frame inference speed of CLIP-L baseline. This proves that the additional computational cost of high resolution is effectively addressed by the efficiency of our sparse tokenization and optimized attention kernel.

---

> ### Author Response · Authors · 2025-11-22
>
> > **W3**: The training data source is limited, relying mainly on the VBench dataset, which may introduce hidden biases.
>
> While VBench is our primary source, it aggregates generated videos from 15 models across different time periods, ensuring high diversity in generative artifacts. To further rule out potential bias, we have worked our best to mitigate the risk of source leakage via data construction and preprocessing:
>   - **Statistical Alignment:** During dataset construction, we carefully balanced the real and synthetic subsets to ensure no significant statistical bias existed in terms of semantic topic, video duration, or native resolution.
>   - **Unified Preprocessing Pipeline**: We applied a unified re-encoding protocol to all data. Both real videos and generated videos in the training set were encoded using H.264 codec and preprocessed with our dynamic resolution pipeline. This step effectively removes specific codec signatures, quantization tables, or metadata artifacts that could serve as "shortcuts," ensuring the model focuses solely on visual content.
>   - **Cross-domain evaluation:** Three different test sets reported in our paper differ greatly from the training set. These evaluations are all cross-domain, cross-source, and cross-generator. Our model consistently maintains strong performance across these benchmarks, demonstrating robust generalization.
>
> ---
> > **Q2**: For Magic Videos, how is the "indistinguishable to humans" claim verified?
>
> - We verify the high visual fidelity and "challenging-to-distinguish" nature of Magic Videos through a combination of quantitative consistency metrics, human subjective evaluation, and vision LLM benchmarking.
> 	- Visual Consistency: Our dataset simulates sophisticated real-world threats using reality-style filtering and detailed prompts. Quantitatively, we observe an average **CLIPSIM score of 0.82** between fake and real videos, indicating high semantic consistency and quality.
> 	- **Human Blind Test:** To directly verify human perception, we conducted a blind study with 10 graduate students. As shown in the following table, human observers achieved an average accuracy of **79.9%**. While humans can still detect these videos better than random chance, this performance is notably lower than the near-perfect detection rates seen in earlier generation benchmarks, confirming that these videos present a significant challenge to human observers.
> 	- **Vision LLM Evaluation:** We further benchmarked against state-of-the-art Vision LLMs (GPT-4o, Gemini-2.5-Pro). As shown in Table 1, these models achieve an average accuracy of only 62.9%–65.4%, lagging significantly behind finetuned approach. This indicates that general-purpose VLLMs, without task-specific fine-tuning, are insufficient for effectively detecting such high-fidelity AI-generated videos.
> -
>   | Detector | Wan2.1 | Seaweed | StepVideo | AVG |
>   |---|---|---|---|---|
>   | GPT-4o (zero shot) | 55.6 | 68.6 | 64.5 | 62.9 |
>   | Gemini-2.5-Pro | 54.7 | 69.1 | 72.3 | 65.4 |
>   | Human | 71.9 | **86.7** | **81.0** | 79.9 |
>   | Qwen2.5-ViT (Ours) | **85.1** | 84.7 | 77.7 | **82.5** |
>
> ---
> > **W4&Q3**: Experimental results show significant performance differences across different generators (from 72.26% to 85.12% in Table 2). Have you analyzed the specific reasons for these differences? Are there any specific types of generated content or technical approaches that are blind spots for the current method?
>
> Because some generated subsets share the same real video samples, the differences in accuracy mainly stem from the detection of fake videos. One possible reason is the variation in generative artifacts, such as lower input resolution, which can make the forgery patterns more difficult for the model to detect.
>
> Failure Case Analysis: The model tends to miss detections in scenarios that lack both high-frequency details and distinct semantic context, such as scenes of pure sky or static bird’s-eye landscapes. Technically, if a generative model produces a video that is nearly pixel-identical to a real one, almost all detectors would fail. However, such samples are essentially harmless. Our goal is to exploit the limitations of generative models and identify the imperfections that arise when they attempt to synthesize content that does not exist in the real world.

---

### Official Review · Reviewer_A394 · 2025-10-31

**Soundness:** 3
**Presentation:** 3
**Contribution:** 3
**Rating:** 4
**Confidence:** 5

**Summary:**

This paper addresses the pressing need for effective detection methods for AI-generated videos, which is a highly valuable area of research. The authors provide an in-depth exploration of the current state of research, highlighting its shortcomings, including the development of video generation models, detection of generated images, and existing methods for detecting generated videos. Two major challenges are identified in the detection of AI-generated videos: 1) The fixed-resolution preprocessing through operations like cropping and downsampling leads to information loss and coarse-grained detection; 2) Current detection methods are typically trained on outdated synthetic data sources, which are insufficient for handling videos produced by the latest high-quality generative frameworks.
In response to these issues, the authors construct a high-quality and diverse dataset sourced from state-of-the-art generative models and propose a novel detection framework built upon Qwen2.5-VL, which processes videos in their native spatial resolution and temporal length. This approach preserves crucial forgery artifacts often lost in conventional preprocessing steps, such as resizing or cropping.
Through extensive experiments, the authors demonstrate the effectiveness of their method and reasonably discuss the remaining challenges. Their work significantly advances the field, providing a robust foundation for future AI-generated video detection efforts.

**Strengths:**

1.This paper addresses the significant limitations of existing methods for detecting AI-generated videos, highlighting the urgent need for effective detection strategies in light of the rapid development of video generation technologies.
2. The authors construct a novel, large-scale dataset by utilizing cutting-edge video generation tools, which ensures that the dataset is diverse and high-quality, effectively supporting the proposed detection framework.

**Weaknesses:**

1.The section on "3D Video Patchifying at Native Scale" lacks sufficient novelty. Although the paper claims that the model is trained at native resolution, this approach was already introduced in Qwen2.5-VL [1]. The focus of the method seems to be more on engineering optimization rather than presenting a fundamentally new contribution to the field. Additional exploration of novel techniques or improvements beyond existing methods would strengthen this section.
2.The paper lacks a more detailed analysis of the proposed method, particularly regarding how the model differentiates between real and generated content. While the authors claim the use of native resolution processing, there is little discussion on which specific parts of the video (e.g., temporal inconsistencies, high-frequency artifacts, or motion patterns) the model focuses on to distinguish between real and synthetic videos. A deeper exploration of the key features the model uses to make this distinction would help clarify the strengths of the approach and provide insights into its decision-making process.
3.While the use of dynamic resolution processing significantly enhances performance, it introduces additional computational overhead. This increased complexity may limit the model’s feasibility for real-world deployment, particularly on resource-constrained devices or in scenarios requiring real-time detection.
4.The proposed dataset structure is highly unreasonable; each class in the test set has very few samples, making the statistical results highly likely to be biased.
5. The experiments were insufficient and were not tested on the latest dataset GenVidBench.
[1]Shuai Bai, Keqin Chen, Xuejing Liu, Jialin Wang, Wenbin Ge, Sibo Song, Kai Dang, Peng Wang, Shijie Wang, Jun Tang, Humen Zhong, Yuanzhi Zhu, Mingkun Yang, Zhaohai Li, Jianqiang Wan, Pengfei Wang, Wei Ding, Zheren Fu, Yiheng Xu, Jiabo Ye, Xi Zhang, Tianbao Xie, Zesen Cheng, Hang Zhang, Zhibo Yang, Haiyang Xu, and Junyang Lin. Qwen2.5-vl technical report, 2025.

**Questions:**

see the weakness

---

> ### Author Response · Authors · 2025-11-22
>
> > **W1**: The focus of the method seems to be more on engineering optimization rather than presenting a fundamentally new contribution to the field. Additional exploration of novel techniques or improvements beyond existing methods would strengthen this section.
>
> Thank you for the constructive feedback. We respectfully argue that our contribution lies not in designing a new network topology from scratch, but in identifying a critical bottleneck in current forgery detection pipelines and validating a **paradigm shift from "fixed-size preprocessing" to "native-resolution modeling."**
> We summarize our contribution as follows:
> 1. **Identifying the Bottleneck: Why Native Resolution Matters.** Existing detectors predominantly rely on fixed-size resizing or cropping. Through systematic cross-validation, we demonstrate that these standard preprocessing steps are the primary bottleneck for detecting high-quality AI-generated video.
>     - **Semantic Distortion:** Fixed resizing ignores the diverse aspect ratios of real-world content (unlike the 1:1 bias often found in generative models), introducing geometric distortions. Meanwhile, cropping leads to severe content loss and lack of global context.
>     - **Loss of Artifacts:** Down-sampling operations effectively act as low-pass filters, erasing the subtle, high-frequency artifacts that are crucial for identifying modern deepfakes.
>     - **Inefficiency:** While sliding windows can preserve resolution, they fail to model global dependencies and significantly increase inference costs.
> 2. The choice to build upon the Qwen2.5-VL architecture is a strategic decision to solve specific technical challenges that traditional backbones (e.g., ResNet, ViT) cannot handle efficiently
>     - **Variable Resolution Support:** Traditional models struggle with variable resolutions due to poor extrapolation of positional embeddings and the computational waste of padding within batches.
>     - **Advanced Modeling Capabilities:** Adapting older backbones to handle native resolutions and 3D temporal modeling would require fundamental re-engineering and massive-scale pre-training.
>     - By leveraging the advanced designs of Qwen2.5-VL (e.g., **RoPE** for extrapolation, **NaViT** + **Flash Attention** for efficiency, and **3D patchifying** for unified video modeling), we overcome the engineering barriers. This allows us to focus on the scientific contribution: proving that preserving the original spatial-temporal information is key to robust detection.
> 3. Beyond the method, we contribute a high-quality, multi-generator dataset and Magic Videos benchmark. We believe this work establishes a strong, reproducible baseline for the community. We commit to open-sourcing our dataset, code, and model weights to facilitate future research in AI-generated video detection.
>
> > **W2**: The paper lacks a more detailed analysis of the proposed method, particularly regarding how the model differentiates between real and generated content. A deeper exploration of the key features the model uses to make this distinction would help clarify the strengths of the approach and provide insights into its decision-making process.
>
> We agree that demonstrating what the model learns is crucial for validating our approach. We have conducted the requested attention visualization experiments and would upload the revised manuscript in a few days.
>
> Current effective AIGC detection typically relies on two complementary categories of discriminative features. The visualization results confirm that our native-resolution framework successfully captures both types of cues, justifying the necessity of our architectural choices:
>   - Capturing Low-level Artifacts: The attention maps show strong activation on fine-grained details, such as unnatural text rendering, inconsistent motion blur. Critically, these are the high-frequency features that are typically eroded by standard down-sampling
>   - Capturing High-level Semantics: The model also attends to global anomalies, such as unnatural object deformations (fruit-cutting videos) or distinct generative styles (e.g., "AI-like" lighting), proving it is not limited to local texture features.
>
> Failure Case Analysis. To further validate this, we analyzed failure cases. The model tends to miss detections in scenarios lacking both high-frequency details and distinct semantic context, such as pure blue sky. This paradoxically confirms that the model is indeed searching for specific forensic traces (semantic or textural) rather than overfitting to superficial dataset biases or shortcut learning.

---

> ### Author Response · Authors · 2025-11-22
>
> > **W3**: While the use of dynamic resolution processing significantly enhances performance, it introduces additional computational overhead. This increased complexity may limit the model’s feasibility for real-world deployment, particularly on resource-constrained devices or in scenarios requiring real-time detection.
>
> Thank you for raising the concern about computational overhead. We acknowledge that processing high-resolution inputs typically implies higher costs. However, we would like to clarify that our proposed method achieves superior efficiency compared to baselines due to its systematic architectural optimizations, specifically the dynamic resolution mechanism and Flash Attention integration.
>
> 1. **Training Efficiency (Referencing Appendix Table 11):** As demonstrated in our Appendix (Page23, Table11), although our backbone may have higher parameters or theoretical FLOPs, the actual training cost is lower.
> |Model|Input Resolution|#Params|FLOPs|Peak GPU Mem|Training Time / A100 hrs per epoch|
> |--|--|--|--|--|--|
> |CLIP-L|224 × 224|303.2M|622.6G|129.3GB|9.5h|
> |XCLIP-L|224 × 224|429.2M|650.6G|129.3GB|10.5h|
> |Effort|224 × 224|0.2M/504.6M|623.4G|75.1GB|7.5h|
> |Qwen2.5-ViT (Ours)|224 × 224|668.7M|656G|16.0GB|2.3h|
> |Qwen2.5-ViT (Ours)|dynamic [224p, 448p]|668.7M|656G|37.9GB|7h|
> |Qwen2.5-ViT-Lora (Ours)|dynamic [224p, 448p]|2.6M / 671.3M|656G|27.4GB|5.5h|
> 2. **Inference Speed Comparison (New Experiment):** To further address the concern regarding real-world deployment, we conducted a detailed inference speed comparison (measured on a single A100 GPU). The results are summarized in the table below:
> ||input resolution|frame|inference time|fps|
> |--|--|--|--|--|
> |CLIP-L|224p|1|3m20s|90.9|
> |CLIP-L|224p|8|24m20s|99.6|
> |XCLIP-L|224p|8|26m34s|93.3|
> |Effort|224p|8|25m1s|96.9|
> |Qwen2.5-ViT (Ours)|force 224p|8|5m27s|444.9|
> |Qwen2.5-ViT (Ours)|dynamic [224p, 448p]|8|13m59s|177.2|
>
>     - **Experiment Settings**: All methods are evaluated on GenVideo-val, which contains 19k videos in resolution from 224p to 1280p. For a fair comparison, we use temporal padding for our method to ensure 8-frame input of all videos.
>     - **Observation:** Our method (using 8 frames with dynamic high resolution) is significantly faster than baseline methods with 8-frame input.
>     - **Key Finding:** Surprisingly, our 8-frame inference speed is comparable to the **single-frame inference speed** of CLIP-L baseline. This proves that the additional computational cost of high resolution is effectively addressed by the efficiency of our sparse tokenization and optimized attention mechanism.
>
> 3. **Conclusion:** Therefore, the "native scale" processing in our framework does not hinder deployment. On the contrary, thanks to the elimination of padding redundancy and the integration of Flash Attention 2, our method achieves higher accuracy with high-resolution details while outperforms the computational efficiency of lower-resolution baselines.
>
> > **W4**: The proposed dataset structure is highly unreasonable; each class in the test set has very few samples, making the statistical results highly likely to be biased.
>
> Our goal is to construct a benchmark that balances high quality, high diversity, and evaluation efficiency. Our test set covers 6 cutting-edge video generation systems, and we carefully curated prompts to cover challenging and diverse scenarios. This ensures that even with a smaller number of samples per class, the test set effectively spans the manifold of the generator's capabilities, providing a dense but comprehensive evaluation.
>
> In Addition, we significantly expanded the test set for the Wan-1.3B generator, increasing the volume from 282 samples to 2,000 samples using a wider array of prompts. We re-evaluated our model on this expanded 2k dataset. The recall rate and AP is statistically identical to those on the 282-sample set.
>
> We are working to incorporate latest video generators, such as veo 3.1 and sora2 to provide a more comprehensive evaluation.

---

> ### Author Response · Authors · 2025-11-22
>
> > **W5**: The experiments were insufficient and were not tested on the latest dataset GenVidBench.
> ---
>
> ### GenVidBench Evaluation
> We evaluated our mode on GenVidBench using the checkpoint trained on our original 140k dataset. The results are summarized in the table below
> |genvidbench|MuseV|svd|Cogvideo|Mora|Reall|AVG|
> |--|--|--|--|--|--|--|
> |Qwen2.5-ViT (Ours)|62.37|59.33|95.95|94.38|93.31|81.07|
>
> As shown in the table, our model achieves a strong Mean Accuracy of 81.07%. The model demonstrates excellent zero-shot generalization on CogVideo (95.95%) and Mora (94.38%), confirming that our method effectively learns universal forgery traces common in Text-to-Video generation.
> A performance drop is also observed on MuseV and SVD. We attribute this to two specific characteristics of these Image-to-Video (I2V) models
>   - **Reliance on Real Reference Images:** These models generate videos based on a real reference image. Consequently, the pixel distribution remains highly similar to the real source.
>   - **Short Duration & Sampling Rate:** Videos from these models are extremely short (~1 second). Since our inference pipeline uses a sampling rate of 2 FPS, the model sees only 2-3 frames. Given the high fidelity of the initial frame (which is real), the model lacks sufficient temporal context to detect motion anomalies or accumulating artifacts.
>
> This analysis provides a valuable direction for improvement. We are working to specifically address this challenging I2V scenario in AIGC detection task.
>
> ---
>
> ### Evaluation on DeepTraceReward Benchmark
>
> To further demonstrate generalization against unseen generators, we evaluated our method on the *DeepTraceReward* benchmark [1], which contains 4,335 videos from 7 recent models (including Pika-1.5, Kling-1.5, etc). We compared our Qwen2.5-ViT against leading Multimodal LLMs reported in [1].
>
> -
>   | Method | ACC | Fake ACC | Real ACC |
>   |---|---|---|---|
>   | GPT-5 | 90.7 | 84.6 | 98.8 |
>   | GPT-4.1 | 92.9 | 89.1 | 97.9 |
>   | Gemini 2.5 Pro | 84.3 | 75.7 | 95.8 |
>   | Qwen 2.5 VL 7B | 51.7 | 20.2 | 93.4 |
>   | DeepTraceReward (w/ Qwen2.5 VL 7B) | 74.7 | 55.7 | 100.0 |
>   | Qwen2.5-ViT (Ours) | **97.2** | **96.3** | 98.2 |
>
>
> - **Superior Generalization:** Our model achieves **97.2% accuracy**, significantly outperforming massive foundation models (e.g., GPT-5, Gemini 2.5 Pro) on the binary classification task.
> - **Balanced Detection:** While general-purpose VLMs often struggle with detecting fakes (showing lower Fake ACC), our model demonstrates balanced performance (96.3% Fake ACC vs. 98.2% Real ACC), proving its effectiveness in identifying artifacts from the latest generation engines without overfitting to specific training generators.
>
> [1] Learning Human-Perceived Fakeness in AI-Generated Videos via Multimodal LLMs, arXiv:2509.22646

---

> ### Comment · Reviewer_A394 · 2025-11-26
> **Rebuttal Response**
>
> Thank you for the author's response. However, I still believe that the improvements in methodology are minor adjustments with limited innovation. For example, in the VIDGUARD-R1 work, not only was the VLM improved, but GRPO training was also conducted. In comparison, this work's contributions seem too limited to justify its acceptance.

---

> > ### Author Response · Authors · 2025-11-27
> >
> > We thank the reviewer for highlighting VIDGUARD-R1, which we recognize as a strong representative of the "Reasoning-centric" VLM direction. However, we respectfully argue that our work and VLM-based approaches represent two complementary research paths, rather than directly competing alternatives:
> > 1. Distinct Goals (Perception vs. Reasoning): While VLMs focus on semantic reasoning (explaining why and locating forgeries), our work targets pure visual perception. We conduct in-depth analysis and propose a solution to the fundamental bottleneck of information loss and potential bias caused by fixed-size downsampling, enabling a native-resolution perception framework.
> > 2. Deployment & Efficiency: Our Pure Vision framework removes the heavy LLM decoder. This offers a high-efficiency, hallucination-free solution suitable for large-scale, real-time processing where 7B+ parameter VLMs are often computationally prohibitive.
> > 3. Dataset Contribution: Beyond architecture, we contribute a high-quality dataset covering the latest generators for both training and evaluation. This is critical infrastructure that benefits the entire field, including future VLM research.
> >
> > We believe the community requires both sophisticated reasoning models and robust, efficient perception solutions.

---

### Author Response · Authors · 2025-12-04
**Summary of Rebuttal**

This paper introduces a large-scale, up-to-date dataset and benchmark, along with a dynamic native-resolution framework for AI-generated video detection. Our contributions, including the timeliness of the task and dataset, the well-motivated methodology, and the strong and comprehensive quantitative results, have been positively acknowledged by the reviewers.

The reviewers mainly raised the following **core concerns**:
  - **Lack of visualization analysis.** In response to the questions about the model’s decision-making process, we have conducted saliency visualization experiments in **Figure 4**. The results show that our model focuses on two types of forgery cues: (1) low-level artifacts such as unnatural text rendering and motion blur, and (2) high-level semantic inconsistencies such as abnormal lighting and object deformations. This confirms that our model indeed relies on both semantic and textural forensic traces rather than overfitting to dataset biases.
  - **Lack of fundamental methodology contribution.** We clarify that our contributions are three-fold: (1) a timely and publicly released training dataset and Magic Videos benchmark, (2) an in-depth pre-experiment and analysis of cross-resolution evaluation and identification of the limitations of fixed-resolution downsampling, and (3) a unified native-resolution framework for AIGC detection with better generalization and efficiency.
  - **Lack of efficiency analysis.** We previously included a training-efficiency comparison in the **Appendix** of the original submission. In addition, we now provide an inference-efficiency comparison. Owing to the use of NaViT and FlashAttention, our Qwen2.5-ViT backbone, supporting a higher dynamic input resolution, requires fewer computational resources than the baselines (CLIP-L, XCLIP-L) during both training and inference.

For most of the **remaining concerns**, we provide detailed experiments and explanations, including robustness under video compression (R3, KYH3), evaluation on GenVidBench (R1, A394), DeepTraceReward (R3), additional real-video sources (R3), comparison to 448p baselines (R4, 9HXd), dataset release policy (R2, X5mz), human blind testing, and comparisons against VLMs (R2).

We have **revised the manuscript** to include additional related-work discussion, robustness comparison (Figure 3), saliency visualization (Figure 4), and extended results on DeepTraceReward (Table 6). We observe a performance drop in detecting I2V (image-to-video) content and plan to further improve our method in future work.

Due to the new policy, we have not yet communicated with all reviewers; however, we have already conducted comprehensive experiments and prepared detailed responses to each reviewer’s questions. Below are summaries of our responses to each reviewer:

---

> ### Author Response · Authors · 2025-12-04
> **Summary of Rebuttal (Part 1/4): Reviewer A394 (Initial Rating: 4, Confidence: 5)**
>
> Review A394 claims that our work addresses key limitations in AI-generated video detection and provides a large, diverse, high-quality dataset from cutting-edge video generators to support effective evaluation.
> > **Concern 1: Limited methodological novelty**
>
> - **Solution:** We show that our key contribution is building a up-to-date dataset and benchmark, identifying and analyzing the fixed-downsampling bottleneck and introducing a native-resolution paradigm enabled by Qwen2.5-VL.
> - **Conclusion:** This constitutes a principled methodological shift rather than merely an engineering refinement.
>
> > **Concern 2: Insufficient analysis of model behavior**
>
> - **Solution:** We added saliency visualizations demonstrating that the model detects both low-level artifacts and high-level semantic inconsistencies.
> - **Conclusion:** These results confirm that the model relies on meaningful forensic cues instead of dataset biases.
>
> > **Concern 3: High computational overhead**
>
> - **Solution:** We provide efficiency benchmarks showing our model is faster than baseline 8-frame methods due to a series of optimization such as NaViT and FlashAttention.
> - **Conclusion:** Native-resolution processing is efficient and practical for real-world deployment.
>
> > **Concern 4: Small test-set sample size**
>
> - **Solution:** We expanded the Wan-1.3B subset to 2,000 samples and confirmed stable performance across both sets.
> - **Conclusion:** The benchmark remains statistically reliable despite small per-class sizes.
>
> > **Concern 5: Missing evaluation on GenVidBench**
>
> - **Solution:** We evaluated on GenVidBench and DeepTraceReward and observed strong generalization, surpassing recent foundation models.
> - **Conclusion:** The method generalizes effectively to the latest AIGC systems and validates our approach.

---

> > ### Author Response · Authors · 2025-12-04
> > **Summary of Rebuttal (Part 2/4): Reviewer X5mz (Initial Rating: 6, Confidence: 3)**
> >
> > The reviewer X5mz acknowledges this paper’s comprehensive experiments on multiple public benchmarks and the Magic Videos test set, its dynamic‑resolution framework with good cross‑dataset generalization, its systematic ablation studies supporting the main claims, and its use of architecture optimization to improve efficiency for practical deployment.
> >
> > > **Concern 1: Decision mechanism and potential bias**
> > >
> > - **Solution**: We added attention visualizations and failure-case analysis showing the model attends to both low-level artifacts and high-level semantic anomalies, and fails mainly when neither is present.
> > - **Conclusion**: This indicates the model relies on genuine forensic cues rather than superficial dataset biases or shortcuts.
> >
> > > **Concern 2: Computational efficiency and deployment practicality**
> > >
> > - **Solution**: We reported training and inference benchmarks (on GenVideo-val) showing our Qwen2.5-ViT with dynamic resolution is faster than 8-frame baselines and comparable to single-frame CLIP-L.
> > - **Conclusion**: The method achieves practical efficiency with high-resolution inputs, supporting real-world deployment.
> >
> > > **Concern 3: Training data bias from VBench**
> > >
> > - **Solution**: We used diverse generators in VBench, balanced real vs. synthetic statistics, applied unified re-encoding/preprocessing, and validated on multiple cross-domain test sets (GenVideo, DVF, DeepTraceReward, bdd-100k, TGIF, etc.)
> > - **Conclusion**: Dataset design and cross-domain results suggest the model is not overly dependent on source-specific biases.
> >
> > > **Concern 4: “Indistinguishable to humans” claim for Magic Videos**
> > >
> > - **Solution**: We combined high CLIPSIM scores, a human blind test, and Vision-LLM benchmarks showing that both humans and general VLLMs find these videos challenging.
> > - **Conclusion**: These results support that Magic Videos are high-fidelity and difficult to distinguish from real videos.
> >
> > > **Concern 5: Performance gaps across generators and blind spots**
> > >
> > - **Solution**: We found that performance gaps mainly arise from varying artifact strength across generators, with failures concentrated in scenes lacking both rich texture and semantics.
> > - **Conclusion**: The method is most challenged by near artifact-free, low-information, or highly-compressed videos, while remaining effective where generative imperfections exist.

---

> > > ### Author Response · Authors · 2025-12-04
> > > **Summary of Rebuttal (Part 3/4): Reviewer KYH3 (Initial Rating: 4, Confidence: 5)**
> > >
> > > The reviewer KYH3 acknowledges this paper’s timely focus on closing the gap with 2024–2025 video generators using refreshed data from new models, its clear and well-supported empirical story about resolution, and its clear, well-written presentation.
> > >
> > > > **Concern 1: Contribution mainly seen as engineering consolidation**
> > > >
> > > - **Solution**: We clarify that our core contribution is to identify fixed-size preprocessing as a fundamental bottleneck and to empirically validate a paradigm shift to native-resolution, spatio‑temporal modeling using an architecture (Qwen2.5‑VL) that can efficiently support this regime.
> > > - **Conclusion**: Our work goes beyond architectural engineering by establishing and validating native‑resolution modeling, together with the Magic Videos benchmark, as a new, reproducible baseline paradigm for AI‑generated video detection.
> > >
> > > > **Concern 2: Dataset release and reproducibility**
> > > >
> > > - **Solution**: We will publicly release all generated Magic Videos alongside their text prompts under an academic research license, ensuring that data, labels, and generation conditions are fully accessible.
> > > - **Conclusion**: This guarantees that Magic Videos is reproducible and reusable by the community in a manner consistent with ICLR’s expectations for openness and reproducibility.
> > >
> > > > **Concern 3: Missing discussion of [1] DuB3D [2] L3DE**
> > > >
> > > - **Solution**: We will integrate [1] and [2] into the related‑work section, highlighting GenVidDet/DuB3D and L3DE, and add comparative or complementary analyses in the experiments when feasible.
> > > - **Conclusion**: Positioning our native‑resolution detection framework alongside these large‑scale and 3D‑coherence approaches will clarify our distinct contribution within the latest AIGC video detection literature.
> > >
> > > > **Concern 4: Source bias and generalization**
> > > >
> > > - **Solution**: We mitigate source leakage via balanced construction, unified H.264 encoding for all videos, and comprehensive cross‑domain tests, including new evaluations on BDD100K, TGIF, Ego4D, and InternVid‑10M.
> > > - **Conclusion**: Strong performance on diverse unseen real sources and on DeepTraceReward supports that our model captures general forgery cues rather than dataset‑ or codec‑specific artifacts.
> > >
> > > > **Concern 5: Robustness to real‑world perturbations**
> > > >
> > > - **Solution**: We analyze robustness to standard image perturbations and add controlled experiments with strong H.264 compression and multiple codecs on MovieGen‑2K, comparing against other baselines.
> > > - **Conclusion**: Our native‑resolution detector consistently degrades more gracefully under heavy compression and across codecs, validating its advantage under realistic platform processing.

---

> > > > ### Author Response · Authors · 2025-12-04
> > > > **Summary of Rebuttal (Part 4/4): Reviewer 9HXd (Initial Rating: 6, Confidence: 4)**
> > > >
> > > > The reviewer 9HXd acknowledges this paper’s focus on the important problem of AI-generated video detection, its contribution of a curated Gen-AI video dataset, and its detection framework that improves over baseline methods.
> > > >
> > > > > **Concern 1: Lack of methodological novelty**
> > > > >
> > > > - **Solution**: We clarify that our main contribution is to identify fixed‑size preprocessing as a core bottleneck and to empirically establish native‑resolution spatio‑temporal modeling (via Qwen2.5‑VL) as a more effective paradigm, supported by a new multi‑generator dataset and benchmark.
> > > > - **Conclusion**: Rather than a mere combination of existing components, our work introduces and validates a native‑resolution detection framework and releases Magic Videos as a strong, reproducible baseline for AI‑generated video detection.
> > > >
> > > > > **Concern 2: Artifacts are not clearly defined or visualized**
> > > > >
> > > > - **Solution**: We conduct attention‑based visualizations showing that our native‑resolution model focuses on both low‑level artifacts (e.g., unnatural text, motion blur inconsistencies) and high‑level semantic anomalies (e.g., object deformations, characteristic “AI‑like” styles).
> > > > - **Conclusion**: These visual analyses make the notion of artifacts explicit and demonstrate that the model exploits both fine‑grained and semantic cues rather than relying on vague or implicit patterns.
> > > >
> > > > > **Concern 3: Possibility of removing or masking artifacts to fool the detector**
> > > > >
> > > > - **Solution**: We observe that the detector tends to fail on scenes lacking both rich details and distinctive semantics (e.g., pure sky or static bird’s‑eye views) or under extreme compression, where generative imperfections are largely suppressed.
> > > > - **Conclusion**: While any detector will fail on videos that are effectively indistinguishable from real ones, our method is designed to exploit the current limitations of generative models by detecting residual imperfections that arise when they synthesize complex content.
> > > >
> > > > > **Concern 4: Unfair comparison due to resolution; unclear if gains come from architecture or access to high‑resolution inputs**
> > > > >
> > > > - **Solution**: We retrofit baselines (NPR, CLIP‑L, XCLIP‑L) to train and infer at higher resolution (448p) by interpolating positional embeddings, and compare them against our dynamic‑resolution model across Magic Videos (high‑res) and Genvideo (low‑res).
> > > > - **Conclusion**: Although higher resolution modestly improves baselines on high‑res data while hurting low‑res performance, our dynamic native‑resolution architecture still achieves the best average accuracy, showing that the gains stem from the resolution‑aware design rather than simply from seeing higher‑resolution inputs.

---

> ### Author Response · Authors · 2025-12-04
> **Appreciation for the Reviewers' Valuable Feedback**
>
> We sincerely thank all reviewers for their constructive feedback and for recognizing our study. We would also like to invite the reviewers to look over the responses and revised manuscript. We hope these revisions provide a more comprehensive evaluation and a deeper understanding of our paper.

---

### Meta-Review · Area_Chair_gXNj · 2026-01-06

**Summary:**

The paper addresses generated video detection highlighting tow weaknesses is existing work: rescaling pre-processing might remove artifacts from the generation process and models are usually trained on a limited amount of often outdated data. Besides a thorough analysis, the paper provides a large representative dataset and trains a model using 3D patchification on the videos original scale.
The reviews were originally borderline, highlighting the importance of the task and the solid initial analysis on the one hand and the lack of technical novelty on the other hand.
Overall, the analysis provided in the paper is still relevant and the dataset, extended after the rebuttal, seems to be a valid contribution.

**Reviewer Concerns:**

reviewer A394
1: Limited methodological novelty --> not addressed
2: Insufficient analysis of model behavior --> addressed
3: High computational overhead --> addressed
4: Small test-set sample size --> addressed to some extent
5: Missing evaluation on GenVidBench --> addressed
reviewer X5mz
1: Decision mechanism and potential bias --> empirically addressed
2: Computational efficiency and deployment practicality --> addressed
3: Training data bias from VBench --> addressed to some extent
4: “Indistinguishable to humans” claim for Magic Videos --> addressed
5: Performance gaps across generators and blind spots --> partly addressed.
reviewer KYH3
1: engineering consolidation --> not addressed.
2: Dataset release and reproducibility --> addressed
3: Missing discussion of DuB3D and L3DE --> not addressed but promised to address in the final version. The authors should indeed include these results!
4: Source bias and generalization --> reasonably addressed
5: Robustness to real‑world perturbations --> addressed
 reviewer 9HXd
1: Lack of methodological novelty --> not addressed
2: Artifact visualization --> addressed
3: Possibility of removing or masking artifacts to fool the detector --> addressed by discussion
4: Unfair comparison due to resolution; unclear if gains come from architecture or access to high‑resolution inputs --> addressed to some extent

**Reviewer Scores:**

The reviews consistently provide borderline scores. After the rebuttal, potentially critical questions regarding biases and the validity of results are resolved to my understanding. The method itself is not technically intricate but in conjunction with the provided analysis and dataset, the paper still makes interesting contributions. I would expect the paper to end up with borderline accept scores (6).

---

### Decision · Program_Chairs · 2026-01-26

Accept (Poster)